# Distributed Newton Can Communicate Less and Resist Byzantine Workers

**Avishek Ghosh**
Department of EECS, UC Berkeley
Berkeley, CA 94720
avishek_ghosh@berkeley.edu

**Raj Kumar Maity**
College of Information and Computer Sciences
UMass Amherst, MA-01002
rajkmaity@cs.umass.edu

**Arya Mazumdar**
College of Information and Computer Sciences
UMass Amherst, MA-01002
arya@cs.umass.edu

## Abstract

We develop a distributed second order optimization algorithm that is communication-efficient as well as robust against Byzantine failures of the worker machines. We propose COMRADE (COMunication-efficient and Robust Approximate Distributed nEwton), an iterative second order algorithm, where the worker machines communicate *only once* per iteration with the center machine. This is in sharp contrast with the state-of-the-art distributed second order algorithms like GIANT [31] and DINGO[6], where the worker machines send (functions of) local gradient and Hessian sequentially; thus ending up communicating twice with the center machine per iteration. Moreover, we show that the worker machines can further compress the local information before sending it to the center. In addition, we employ a simple norm based thresholding rule to filter-out the Byzantine worker machines. We establish the linear-quadratic rate of convergence of COMRADE and establish that the communication savings and Byzantine resilience result in only a small statistical error rate for arbitrary convex loss functions. To the best of our knowledge, this is the first work that addresses the issue of Byzantine resilience in second order distributed optimization. Furthermore, we validate our theoretical results with extensive experiments on synthetic and benchmark LIBSVM [4] data-sets and demonstrate convergence guarantees.

## 1   Introduction

In modern data-intensive applications like image recognition, conversational AI and recommendation systems, the size of training datasets has grown in such proportions that distributed computing have become an integral part of machine learning. To this end, a fairly common distributed learning framework, namely *data parallelism*, distributes the (huge) data-sets over multiple *worker machines* to exploit the power of parallel computing. In many applications, such as Federated Learning [17], data is stored in users' personal devices and judicious exploitation of the on-device machine intelligence can speed up computation. Usually, in a distributed learning framework, computation (such as processing, training) happens in the worker machines and the local results are communicated to a *center machine* (ex., a parameter server). The center machine updates the model parameters by properly aggregating the local results.

Such distributed frameworks face the following two fundamental challenges: First, the parallelism gains are often bottle-necked by the heavy communication overheads between worker and the center machines. This issue is further exacerbated where large clusters of worker machines are used for

modern deep learning applications using models with millions of parameters (NLP models, such as BERT [9], may have well over 100 million parameters). Furthermore, in Federated Learning, this uplink cost is tied to the user's upload bandwidth. Second, the worker machines might be susceptible to errors owing to data crashes, software or hardware bugs, stalled computation or even malicious and co-ordinated attacks. This inherent unpredictable (and potentially adversarial) nature of worker machines is typically modeled as Byzantine failures. As shown in [18], Byzantine behavior a single worker machine can be fatal to the learning algorithm.

Both these challenges, communication efficiency and Byzantine-robustness, have been addressed in a significant number of recent works, albeit mostly separately. For communication efficiency, several recent works [29, 27, 2, 13, 1, 32, 16] use quantization or sparsification schemes to compress the message sent by the worker machines to the center machine. An alternative, and perhaps more natural way to reduce the communication cost (via reducing the number of iterations) is to use second order optimization algorithms; which are known to converge much faster than their first order counterparts. Indeed, a handful of algorithms has been developed using this philosophy, such as DANE [24], DISCO [35], GIANT [31], DINGO [6], Newton-MR [23], INEXACT DANE and AIDE [22]. On the other hand, the problem of developing Byzantine-robust distributed algorithms has also been considered recently (see [26, 12, 5, 33, 34, 14, 3] ). However, *all* of these papers analyze different variations of the gradient descent, the standard first order optimization algorithm.

In this work, we propose COMRADE, a distributed approximate Newton-type algorithm that communicates less and is resilient to Byzantine workers. Specifically, we consider a distributed setup with $m$ worker machines and one center machine. The goal is to minimize a regularized convex loss $f : \mathbb{R}^d \to \mathbb{R}$, which is additive over the available data points. Furthermore, we assume that $\alpha$ fraction of the worker machines are Byzantine, where $\alpha \in [0, 1/2)$. We assume that Byzantine workers can send any arbitrary values to the center machine. In addition, they may completely know the learning algorithm and are allowed to collude with each other. To the best of our knowledge, this is the first paper that addresses the problem of Byzantine resilience in second order optimization.

In our proposed algorithm, the worker machines communicate *only once* per iteration with the center machine. This is in sharp contrast with the state-of-the-art distributed second order algorithms (like GIANT [31], DINGO [6], Determinantal Averaging [8]), which sequentially estimates functions of local gradients and Hessians and communicate them with the center machine. In this way, they end up communicating twice per iteration with the center machine. We show that this sequential estimation is redundant. Instead, in COMRADE, the worker machines only send a $d$ dimensional vector, the product of the inverse of local Hessian and the local gradient. Via sketching arguments, we show that the empirical mean of the product of local Hessian inverse and local gradient is close to the global Hessian inverse and gradient product, and thus just sending the above-mentioned product is sufficient to ensure convergence. Hence, in this way, we save $\mathcal{O}(d)$ bits of communication per iteration. Furthermore, in Section 5, we argue that, in order to cut down further communication, the worker machines can even compress the local Hessian inverse and gradient product. Specifically, we use a (generic) $\rho$-approximate compressor ([16]) for this, that encompasses sign-based compressors like QSGD [1] and top$_k$ sparsification [25].

For Byzantine resilience, COMRADE employs a simple thresholding policy on the norms of the local Hessian inverse and local gradient product. Note that norm-based thresholding is computationally much simpler in comparison to existing co-ordinate wise median or trimmed mean ([33]) algorithms. Since the norm of the Hessian-inverse and gradient product determines the *amount* of movement for Newton-type algorithms, this norm corresponds to a natural metric for identifying and filtering out Byzantine workers.

**Our Contributions:** We propose a communication efficient Newton-type algorithm that is robust to Byzantine worker machines. Our proposed algorithm, COMRADE takes as input the local Hessian inverse and gradient product (or a compressed version of it) from the worker machines, and performs a simple thresholding operation on the norm of the said vector to discard $\beta > \alpha$ fraction of workers having largest norm values. We prove the linear-quadratic rate of convergence of our proposed algorithm for strongly convex loss functions. In particular, suppose there are $m$ worker machines, each containing $s$ data points; and let $\boldsymbol{\Delta}_t = \mathbf{w}_t - \mathbf{w}^*$, where $\mathbf{w}_t$ is the $t$-th iterate of COMRADE, and $\mathbf{w}^*$ is the optimal model we want to estimate. In Theorem 2, we show that

$$\|\boldsymbol{\Delta}_{t+1}\| \leq \max\{\Psi_t^{(1)}\|\boldsymbol{\Delta}_t\|, \Psi_t^{(2)}\|\boldsymbol{\Delta}_t\|^2\} + (\Psi_t^{(3)} + \alpha)\sqrt{\frac{1}{s}},$$

where $\{\Psi_t^{(i)}\}_{i=1}^3$ are quantities dependent on several problem parameters. Notice that the above implies a quadratic rate of convergence when $\|\boldsymbol{\Delta}_t\| \geq \Psi_t^{(1)}/\Psi_t^{(2)}$. Subsequently, when $\|\boldsymbol{\Delta}_t\|$ becomes sufficiently small, the above condition is violated and the convergence slows down to a linear rate. The error-floor, which is $\mathcal{O}(1/\sqrt{s})$ comes from the Byzantine resilience subroutine in conjunction with the simultaneous estimation of Hessian and gradient. Furthermore, in Section 5, we consider worker machines compressing the local Hessian inverse and gradient product via a $\rho$-approximate compressor [16], and show that the (order-wise) rate of convergence remain unchanged, and the compression factor, $\rho$ affects the constants only.

We experimentally validate our proposed algorithm, COMRADE, with several benchmark data-sets. We consider several types of Byzantine attacks and observe that COMRADE is robust against Byzantine worker machines, yielding better classification accuracy compared to the existing state-of-the-art second order algorithms.

A major technical challenge of this paper is to approximate local gradient and Hessian simultaneously in the presence of Byzantine workers. We use sketching, similar to [31], along with the norm based Byzantine resilience technique. Using *incoherence* (defined shortly) of the local Hessian along with concentration results originating from uniform sampling, we obtain the simultaneous gradient and Hessian approximation. Furthermore, ensuring at least one non-Byzantine machine gets trimmed at every iteration of COMRADE, we control the influence of Byzantine workers.

**Related Work:** *Second order Optimization:* Second order optimization has received a lot of attention in the recent years in the distributed setting owing to its attractive convergence speed. The fundamentals of second order optimization is laid out in [24], and an extension with better convergence rates is presented in [22]. Recently, in GIANT [31] algorithm, each worker machine computes an approximate Newton direction in each iteration and the center machine averages them to obtain a *globally improved* approximate Newton direction. Furthermore, DINGO [6] generalizes second order optimization beyond convex functions by extending the Newton-MR [23] algorithm in a distributed setting. Very recently, [8] proposes Determinantal averaging to correct the inversion bias of the second order optimization. A slightly different line of work ([30], [15], [21]) uses Hessian sketching to solve a large-scale distributed learning problems.

*Byzantine Robust Optimization:* In the seminal work of [12], a generic framework of one shot median based robust learning has been proposed and analyzed in the distributed setting. The issue of Byzantine failure is tackled by grouping the servers in batches and computing the median of batched servers in [5] (the median of means algorithm). Later in [33, 34], co-ordinate wise median, trimmed mean and iterative filtering based algorithm have been proposed and optimal statistical error rate is obtained. Also, [20, 7] consider adversaries may steer convergence to bad local minimizers for non-convex optimization problems.

**Organization:** In Section 3, we first analyze COMRADE with *one round* of communication per iteration. We assume $\alpha = 0$, and focus on the communication efficiency aspect only. Subsequently, in Section 4, we make $\alpha \neq 0$, thereby addressing communication efficiency and Byzantine resilience simultaneously. Further, in Section 5 we augment a compression scheme along with the setting of Section 4. Finally, in Section 6, we validate our theoretical findings with experiments. Proofs of all theoretical results can be found in the supplementary material.

**Notation:** For a positive integer $r$, $[r]$ denotes the set $\{1, 2, \ldots, r\}$. For a vector $v$, we use $\|v\|$ to denote the $\ell_2$ norm unless otherwise specified. For a matrix $X$, we denote $\|X\|_2$ denotes the operator norm, $\sigma_{max}(X)$ and $\sigma_{min}(X)$ denote the maximum and minimum singular value. Throughout the paper, we use $C, C_1, c, c_1$ to denote positive universal constants, whose value changes with instances.

## 2 Problem Formulation

We begin with the standard statistical learning framework for empirical risk minimization, where the objective is to minimize the following loss function:

$$f(\mathbf{w}) = \frac{1}{n} \sum_{j=1}^{n} \ell_j(\mathbf{w}^T \mathbf{x}_j) + \frac{\lambda}{2} \|\mathbf{w}\|^2, \tag{1}$$

where, the loss functions $\ell_j : \mathbb{R} \to \mathbb{R}$, $j \in [n]$ are *convex, twice differentiable and smooth*. Moreover, $\mathbf{x}_1, \mathbf{x}_2, \ldots, \mathbf{x}_n \in \mathbb{R}^d$ denote the input feature vectors and $y_1, y_2, \ldots, y_n \in \mathbb{R}$ denote the correspond-

ing responses. Furthermore, we assume that the function $f$ is *strongly convex*, implying the existence of a unique minimizer of (1). We denote this minimizer by $\mathbf{w}^*$. Note that the response $\{y_j\}_{j=1}^n$ is captured by the corresponding loss function $\{\ell_j\}_{j=1}^n$. Some examples of $\ell_j$ are

$$\text{logistic loss: } \ell_j(z_j) = \log(1 - \exp(-z_j y_j)), \quad \text{squared loss: } \ell_j(z_j) = \frac{1}{2}(z_j - y_j)^2$$

We consider the framework of distributed optimization with $m$ worker machines, where the feature vectors and the loss functions $(\mathbf{x}_1, \ell_1), \ldots, (\mathbf{x}_n, \ell_n)$ are partitioned homogeneously among them. Furthermore, we assume that $\alpha$ fraction of the worker machines are Byzantine for some $\alpha < \frac{1}{2}$. The Byzantine machines, by nature, may send any arbitrary values to the center machine. Moreover, they can even collude with each other and plan malicious attacks with complete information of the learning algorithm.

## 3  COMRADE Can Communicate Less

We first present the Newton-type learning algorithm, namely COMRADE without any Byzantine workers, i.e., $\alpha = 0$. It is formally given in Algorithm 1 (with $\beta = 0$). In each iteration of our algorithm, every worker machine computes the local Hessian and local gradient and sends the local second order update (which is the product of the inverse of the local Hessian and local gradient) to the center machine. The center machine aggregates the updates from the worker machines by averaging them and updates the model parameter $\mathbf{w}$. Later the center machine broadcast the parameter $\mathbf{w}$ to all the worker machines.

In any iteration $t$, a standard Newton algorithm requires the computation of exact Hessian ($\mathbf{H}_t$) and gradient ($\mathbf{g}_t$) of the loss function which can be written as

$$\mathbf{g}_t = \frac{1}{n}\sum_{i=1}^n \ell_j'(\mathbf{w}_t^\top \mathbf{x}_i)\mathbf{x}_i + \lambda \mathbf{w}_t, \ \ \mathbf{H}_t = \frac{1}{n}\sum_{i=1}^n \ell_j''(\mathbf{w}_t^\top \mathbf{x}_i)\mathbf{x}_i\mathbf{x}_i^\top + \lambda \mathbf{I}. \tag{2}$$

In a distributed set up, the exact Hessian ($\mathbf{H}_t$) and gradient ($\mathbf{g}_t$) can be computed in parallel in the following manner. In each iteration, the center machine 'broadcasts' the model parameter $\mathbf{w}_t$ to the worker machines and each worker machine computes its own local gradient and Hessian. Then the center machine can compute the exact gradient and exact Hessian by averaging the the local gradient vectors and local Hessian matrices. But for each worker machine the per iteration communication complexity is $\mathcal{O}(d)$ for the gradient computation and $\mathcal{O}(d^2)$ for the Hessian computation. Using Algorithm 1, we reduce the communication cost to only $\mathcal{O}(d)$ per iteration, which is the same as the first order methods.

Each worker machine possess $s$ samples drawn uniformly from $\{(\mathbf{x}_1, \ell_1), (\mathbf{x}_2, \ell_2), \ldots, (\mathbf{x}_n, \ell_n)\}$. By $S_i$, we denote the indices of the samples held by worker machine $i$. At any iteration $t$, the worker machine computes the local Hessian $\mathbf{H}_{i,t}$ and local gradient $\mathbf{g}_{i,t}$ as

$$\mathbf{g}_{i,t} = \frac{1}{s}\sum_{i \in S_i} \ell_j'(\mathbf{w}_t^\top \mathbf{x}_i)\mathbf{x}_i + \lambda \mathbf{w}_t, \quad \mathbf{H}_{i,t} = \frac{1}{s}\sum_{i \in S_i} \ell_j''(\mathbf{w}_t^\top \mathbf{x}_i)\mathbf{x}_i\mathbf{x}_i^\top + \lambda \mathbf{I}. \tag{3}$$

It is evident from the uniform sampling that $\mathbb{E}[\mathbf{g}_{i,t}] = \mathbf{g}_t$ and $\mathbb{E}[\mathbf{H}_{i,t}] = \mathbf{H}_t$. The update direction from the worker machine is defined as $\hat{\mathbf{p}}_{i,t} = (\mathbf{H}_{i,t})^{-1}\mathbf{g}_{i,t}$. Each worker machine requires $O(sd^2)$ operations to compute the Hessian matrix $\mathbf{H}_{i,t}$ and $O(d^3)$ operations to invert the matrix. In practice, the computational cost can be reduced by employing conjugate gradient method. The center machine computes the parameter update direction $\hat{\mathbf{p}}_t = \frac{1}{m}\sum_{i=1}^m \hat{\mathbf{p}}_{i,t}$.

We show that given large enough sample in each worker machine ($s$ is large) and with incoherent data points (the information is spread out and not concentrated to a small number of sample data points), the local Hessian $\mathbf{H}_{i,t}$ is close to the global Hessian $\mathbf{H}_t$ in spectral norm, and the local gradient $\mathbf{g}_{i,t}$ is close to the global gradient $\mathbf{g}_t$. Subsequently, we prove that the empirical average of the local updates acts as a good proxy for the global Newton update and achieves good convergence guarantee.

**Algorithm 1** COMmunication-efficient and Robust Approximate Distributed nEwton (COMRADE)

1: **Input:** Step size $\gamma$, parameter $\beta \geq 0$
2: **Initialize:** Initial iterate $w_0 \in \mathbb{R}^d$
3: **for** $t = 0, 1, \ldots, T-1$ **do**
4:     Central machine: broadcasts $w_t$
        **for** $i \in [m]$ **do in parallel**
5:     $i$-th worker machine:
            • Non-Byzantine: Computes local gradient $\mathbf{g}_{i,t}$ and local Hessian $\mathbf{H}_{i,t}$; sends $\hat{\mathbf{p}}_{i,t} = (\mathbf{H}_{i,t})^{-1}\mathbf{g}_{i,t}$ to the central machine,
            • Byzantine: Generates $\star$ (arbitrary), and sends it to the center machine
        **end for**
6:     Center Machine:
            • Sort the worker machines in a non decreasing order according to norm of updates $\{\hat{\mathbf{p}}_{i,t}\}_{i=1}^m$ from the local machines
            • Return the indices of the first $1 - \beta$ fraction of machines as $\mathcal{U}_t$,
            • Approximate Newton Update direction : $\hat{\mathbf{p}}_t = \frac{1}{|\mathcal{U}_t|} \sum_{i \in \mathcal{U}_t} \hat{\mathbf{p}}_{i,t}$
            • Update model parameter: $w_{t+1} = w_t - \gamma \hat{\mathbf{p}}_t$.
7: **end for**

## 3.1 Theoretical Guarantee

We define the matrix $\mathbf{A}_t^\top = [\mathbf{a}_1^\top, \ldots, \mathbf{a}_n^\top] \in \mathbb{R}^{d \times n}$ where $\mathbf{a}_j = \sqrt{\ell_j''(\mathbf{w}^\top \mathbf{x}_j)}\, \mathbf{x}_j$. So the exact Hessian in equation (2) is $\mathbf{H}_t = \frac{1}{n}\mathbf{A}_t^\top \mathbf{A}_t + \lambda \mathbf{I}$. Also we define $\mathbf{B}_t = [\mathbf{b}_1, \ldots, \mathbf{b}_n] \in \mathbb{R}^{d \times n}$ where $\mathbf{b}_i = \ell_i'(\mathbf{w}^T \mathbf{x}_i)\mathbf{x}_i$. So the exact gradient in equation (2) is $\mathbf{g}_t = \frac{1}{n}\mathbf{B}_t\mathbf{1} + \lambda \mathbf{w}_t$

**Definition 1** (Coherence of a Matrix). *Let $\mathbf{A} \in \mathbb{R}^{n \times d}$ be any matrix with $\mathbf{U} \in \mathbb{R}^{n \times d}$ being its orthonormal basis (the left singular vectors). The row coherence of the matrix $\mathbf{A}$ is defined as $\mu(\mathbf{A}) = \frac{n}{d} \max_i \|\mathbf{u}_i\|^2 \in \left[1, \frac{n}{d}\right]$, where $\mathbf{u}_i$ is the $i$th row of $\mathbf{U}$.*

**Remark 1.** *If the coherence of $\mathbf{A}_t$ is small, it can be shown that the Hessian matrix can be approximated well via selecting a subset of rows. Note that this is a fairly common to use coherence condition as an approximation tool (see [10, 11, 19])*

In the following, we assume that the Hessian matrix is $L$-Lipschitz (see definition below), which is a standard assumption for the analysis of the second order method for general smooth loss function (as seen in [31],[8]).

**Assumption 1.** *The Hessian matrix of the loss function $f$ is $L$-Lipschitz continuous i.e. $\left\|\nabla^2 f(w) - \nabla^2 f(w')\right\|_2 \leq L \|w - w'\|$.*

In the following theorem, we provide the convergence rate of COMRADE (with $\alpha = \beta = 0$) in the terms of $\mathbf{\Delta}_t = \mathbf{w}_t - \mathbf{w}^*$. Also, we define $\kappa_t = \sigma_{max}(\mathbf{H}_t)/\sigma_{min}(\mathbf{H}_t)$ as the condition number of $\mathbf{H}_t$, and hence $\kappa_t \geq 1$.

**Theorem 1.** *Let $\mu \in \left[1, \frac{n}{d}\right]$ be the coherence of $\mathbf{A}_t$. Suppose $\gamma = 1$ and $s \geq \frac{3\mu d}{\eta^2} \log \frac{md}{\delta}$ for some $\eta, \delta \in (0, 1)$. Under Assumption 1, with probability exceeding $1 - \delta$, we obtain*

$$\|\mathbf{\Delta}_{t+1}\| \leq \max\left\{\sqrt{\kappa_t\left(\frac{\zeta^2}{1 - \zeta^2}\right)}\|\mathbf{\Delta}_t\|, \frac{L}{\sigma_{min}(\mathbf{H}_t)}\|\mathbf{\Delta}_t\|^2\right\} + \frac{2\epsilon}{\sqrt{\sigma_{min}(\mathbf{H}_t)}},$$

*where $\zeta = \nu(\frac{\eta}{\sqrt{m}} + \frac{\eta^2}{1-\eta})$, $\nu = \frac{\sigma_{max}(\mathbf{A}^\top \mathbf{A})}{\sigma_{max}(\mathbf{A}^\top \mathbf{A}) + n\lambda} \leq 1$, and*

$$\epsilon = \frac{1}{1-\eta} \frac{1}{\sqrt{\sigma_{min}(\mathbf{H}_t)}}(1 + \sqrt{2\ln(\frac{m}{\delta})})\sqrt{\frac{1}{s}} \max_i \|\mathbf{b}_i\|. \qquad (4)$$

**Remark 2.** *It is well known that a distributed Newton method has linear-quadratic convergence rate. In Theorem 1 the quadratic term comes from the standard analysis of Newton method. The linear term (which is small) arises owing to Hessian approximation. It gets smaller with better Hessian approximation (smaller $\eta$), and thus the above rate becomes quadratic one. The small error floor*

*arises due to the gradient approximation in the worker machines, which is essential for the one round of communication per iteration. The error floor is $\propto \frac{1}{\sqrt{s}}$ where $s$ is the number of samples in each worker machine. So for a sufficiently large $s$, the error floor becomes negligible.*

**Remark 3.** *The sample size in each worker machine is dependent on the coherence of the matrix $\mathbf{A}_t$ and the dimension $d$ of the problem. Theoretically, the analysis is feasible for the case of $s \geq d$ (since we work with $\mathbf{H}_{i,t}^{-1}$). However, when $s < d$, one can replace the inverse by a pseudo-inverse (modulo some changes in convergence rate).*

# 4   COMRADE Can Resist Byzantine Workers

In this section, we analyze COMRADE with Byzantine workers. We assume that $\alpha(< 1/2)$ fraction of worker machines are Byzantine. We define the set of Byzantine worker machines by $\mathcal{B}$ and the set of the good (non-Byzantine) machines by $\mathcal{M}$. COMRADE employs a 'norm based thresholding' scheme on the local Hessian inverse and gradient product to tackle the Byzantine workers.

In the $t$-th iteration, the center machine outputs a set $\mathcal{U}_t$ with $|\mathcal{U}_t| = (1-\beta)m$, consisting the indices of the worker machines with smallest norm. Hence, we 'trim' the worker machines that may try to diverge the learning algorithm. We denote the set of trimmed machines as $\mathcal{T}_t$. Moreover, we take $\beta > \alpha$ to ensure at least one good machine falls in $\mathcal{T}_t$. This condition helps us to control the Byzantine worker machines. Finally, the update is given by $\hat{\mathbf{p}}_t = \frac{1}{|\mathcal{U}_t|} \sum_{i \in \mathcal{U}_t} \hat{\mathbf{p}}_{i,t}$. We define:

$$\epsilon_{byz}^2 = [3(\frac{1-\alpha}{1-\beta})^2 + 4\kappa_t(\frac{\alpha}{1-\beta})^2]\epsilon^2, \tag{5}$$

$$\zeta_{byz}^2 = 2(\frac{1-\alpha}{1-\beta})^2(\frac{\nu}{1-\eta})^2 + \nu^2(\frac{1-\alpha}{1-\beta})^2(\frac{\eta}{\sqrt{(1-\alpha)m}} + \frac{\eta^2}{1-\eta})^2 + 4\kappa_t(\frac{\alpha}{1-\beta})^2[2 + (\frac{\nu}{1-\eta})^2]. \tag{6}$$

$\epsilon$ is defined in (4), $\nu = \frac{\sigma_{max}(\mathbf{A}^T\mathbf{A})}{\sigma_{max}(\mathbf{A}^T\mathbf{A})+n\lambda}$ and $\kappa_t$ is the condition number of the exact Hessian $\mathbf{H}_t$.

**Theorem 2.** *Let $\mu \in \left[1, \frac{n}{d}\right]$ be the coherence of $\mathbf{A}_t$. Suppose $\gamma = 1$ and $s \geq \frac{3\mu d}{\eta^2} \log \frac{md}{\delta}$ for some $\eta, \delta \in (0,1)$. For $0 \leq \alpha < \beta < 1/2$, under Assumption 1, with probability exceeding $1 - \delta$, Algorithm 1 yields*

$$\|\mathbf{\Delta}_{t+1}\| \leq \max\{\sqrt{\kappa_t(\frac{\zeta_{byz}^2}{1-\zeta_{byz}^2})}\|\mathbf{\Delta}_t\|, \frac{L}{\sigma_{min}(\mathbf{H}_t)}\|\mathbf{\Delta}_t\|^2\} + \frac{2\epsilon_{byz}}{\sqrt{\sigma_{min}(\mathbf{H}_t)}},$$

*where $\zeta_{byz}$ and $\epsilon_{byz}$ are defined in equations (5) and (6) respectively.*

The remarks of Section 3 is also applicable here. On top of that, we have the following remarks:

**Remark 4.** *Compared to the convergence rate of Theorem 1, the rate here remains order-wise same even with Byzantine robustness. The coefficient of the quadratic term remains unchanged but the linear rate and the error floor suffers a little bit (by a small constant factor).*

**Remark 5.** *Note that for Theorem 2 to hold, we require $\alpha \sim 1/\sqrt{\kappa_t}$ for all $t$. In cases where $\kappa_t$ is large, this can impose a stricter condition on $\alpha$. However, we conjecture that this dependence can be improved via applying a more intricate (and perhaps computation heavy) Byzantine resilience algorithm. In this work, we kept the Byzantine resilience scheme simple at the expense of this condition on $\alpha$.*

# 5   COMRADE Can Communicate Even Less and Resist Byzantine Workers

In Section 3 we analyze COMRADE with an additional feature. We let the worker machines further reduce the communication cost by applying a generic class of $\rho$-approximate compressor [16] on the parameter update of Algorithm 1. We first define the class of $\rho$-approximate compressor:

**Definition 2.** *An operator $\mathcal{Q} : \mathbb{R}^d \to \mathbb{R}^d$ is defined as $\rho$-approximate compressor on a set $S \subset \mathbb{R}^d$ if, $\forall x \in S, \|\mathcal{Q}(x) - x\|^2 \leq (1-\rho) \|x\|^2$, where $\rho \in [0,1]$ is the compression factor.*

The above definition can be extended for any randomized operator $\mathcal{Q}$ satisfying $\mathbb{E}(\|\mathcal{Q}(x) - x\|^2) \leq (1-\rho) \|x\|^2$, for all $\forall x \in S$. The expectation is taken over the randomization of the operator. Notice

that $\rho = 1$ implies that $\mathcal{Q}(x) = x$ (no compression). Examples of $\rho$-approximate compressor include QSGD [1], $\ell_1$-QSGD [16], $top_k$ sparsification and $rand_k$ [25].

Worker machine $i$ computes the product of local Hessian inverse inverse and local gradient and then apply $\rho$-approximate compressor to obtain $\mathcal{Q}(\mathbf{H}_{i,t}^{-1}\mathbf{g}_{i,t})$; and finally sends this compressed vector to the center. The Byzantine resilience subroutine remains the same–except, instead of sorting with respect to $\|\mathbf{H}_{i,t}^{-1}\mathbf{g}_{i,t}\|$, the center machine now sorts according to $\|\mathcal{Q}(\mathbf{H}_{i,t}^{-1}\mathbf{g}_{i,t})\|$. The center machine aggregates the compressed updates by averaging $\mathcal{Q}(\hat{\mathbf{p}}) = \frac{1}{|\mathcal{U}_t|}\sum_{i \in \mathcal{U}_t}\mathcal{Q}(\hat{\mathbf{p}}_{i,t})$, and take the next step as $w_{t+1} = w_t - \gamma\mathcal{Q}(\hat{\mathbf{p}})$.

Recall the definition of $\epsilon$ from (4). We also use the following notation : $\zeta_{\mathcal{M}}^2 = \nu(\frac{\eta}{\sqrt{(1-\alpha)m}} + \frac{\eta^2}{1-\eta}), \zeta_1 = \frac{\nu}{1-\eta}$ and $\nu = \frac{\sigma_{max}(\mathbf{A}^T\mathbf{A})}{\sigma_{max}(\mathbf{A}^T\mathbf{A})+n\lambda}$. Furthermore, we define the following:

$$\epsilon_{comp,byz}^2 = [3(\frac{1-\alpha}{1-\beta})^2 + 4\kappa_t(\frac{\alpha}{1-\beta})^2](1 + \kappa(1-\rho))\epsilon^2 \tag{7}$$

$$\zeta_{comp,byz}^2 = 2(\frac{1-\alpha}{1-\beta})^2(\zeta_1^2 + \kappa_t(1-\rho)((1+\zeta_1^2)) + (\frac{1-\alpha}{1-\beta})^2(\zeta_{\mathcal{M}}^2 + \kappa_t(1-\rho)((1+\zeta_1^2))$$
$$+ 4\kappa_t(\frac{\alpha}{1-\beta})^2(2 + (\zeta_1^2 + \kappa_t(1-\rho)((1+\zeta_1^2)))) \tag{8}$$

**Theorem 3.** *Let $\mu \in \left[1, \frac{n}{d}\right]$ be the coherence of $\mathbf{A}_t$ . Let $\gamma = 1$ and $s \geq \frac{3\mu d}{\eta^2}\log\frac{md}{\delta}$ for some $\eta, \delta \in (0,1)$. For $0 \leq \alpha < \beta < 1/2$, under Assumption 1 and with $\mathcal{Q}$ being the $\rho$-approximate compressor, with probability exceeding $1 - \delta$, we obtain*

$$\|\mathbf{\Delta}_{t+1}\| \leq \max\{\sqrt{\kappa_t(\frac{\zeta_{comp,byz}^2}{1-\zeta_{comp,byz}^2})}\|\mathbf{\Delta}_t\|, \frac{L}{\sigma_{min}(\mathbf{H}_t)}\|\mathbf{\Delta}_t\|^2\} + \frac{\epsilon_{comp,byz}}{\sqrt{\sigma_{min}(\mathbf{H}_t)}}$$

*where $\epsilon_{comp,byz}$ and $\zeta_{comp,byz}$ are given in equations (7) and (8) respectively.*

**Remark 6.** *With no compression ($\rho = 1$) we get back the convergence guarantee of Theorem 2.*

**Remark 7.** *Note that even with compression, we retain the linear-quadratic rate of convergence of COMRADE. The constants are affected by a $\rho$-dependent term.*

## 6 Experimental Results

In this section we validate our algorithm, COMRADE in Byzantine and non-Byzantine setup on synthetically generated and benchmark LIBSVM [4] data-set. The experiments focus on the standard logistic regression problem. The logistic regression objective is defined as $\frac{1}{n}\sum_{i=1}^n \log\left(1 + \exp(-y_i\mathbf{x}_i^\top\mathbf{w})\right) + \frac{\lambda}{2n}\|\mathbf{w}\|^2$, where $\mathbf{w} \in \mathbb{R}^d$ is the parameter, $\{\mathbf{x}_i\}_{i=1}^n \in \mathbb{R}^d$ are the feature data and $\{y_i\}_{i=1}^n \in \{0,1\}$ are the corresponding labels. We use 'mpi4py' package for distributed framework (swarm2) at the University of Massachusetts Amherst [28] using mpi4py Python package. We choose 'a9a' ($d = 123, n \approx 32K$), 'w5a' ($d = 300, n \approx 10k$), 'Epsilon' ($d = 2000, n = 0.4M$) and 'covtype.binary' ($d = 54, n \approx 0.5M$) classification datasets and partition the data in 20 different worker machines. In the experiments, we choose two types of Byzantine attacks : (1). 'flipped label'-attack where (for binary classification) the Byzantine worker machines flip the labels of the data, thus making the model learn with wrong labels, and (2). 'negative update attack' where the Byzantine worker machines compute the local update ($\hat{\mathbf{p}}_i$) and communicate $-c \times \hat{\mathbf{p}}_i$ with $c \in (0,1)$ making the updates to be opposite of actual direction. We choose $\beta = \alpha + \frac{2}{m}$. We choose the regularization parameter $\lambda = 1$ and fixed step size. We ran the algorithms sufficient number of steps to ensure convergence.

In Figure 1(first row) we compare COMRADE in non-Byzantine setup ($\alpha = \beta = 0$) with the state-of the art algorithm GIANT [31]. It is evident from the plot that despite the fact that COMRADE requires less communication, the algorithm is able to achieve similar accuracy. Also, we show the ineffectiveness of GIANT in the presence of Byzantine attacks. In Figure 2((e),(f)) we show the accuracy for flipped label and negative update attacks. These plots are an indicator of the requirement of robustness in the learning algorithm. So we device 'Robust GIANT', which is GIANT algorithm with added 'norm based thresholding' for robustness. In particular, we trim the worker machines based on the local gradient norm in the first round of communication of GIANT. Subsequently, in the

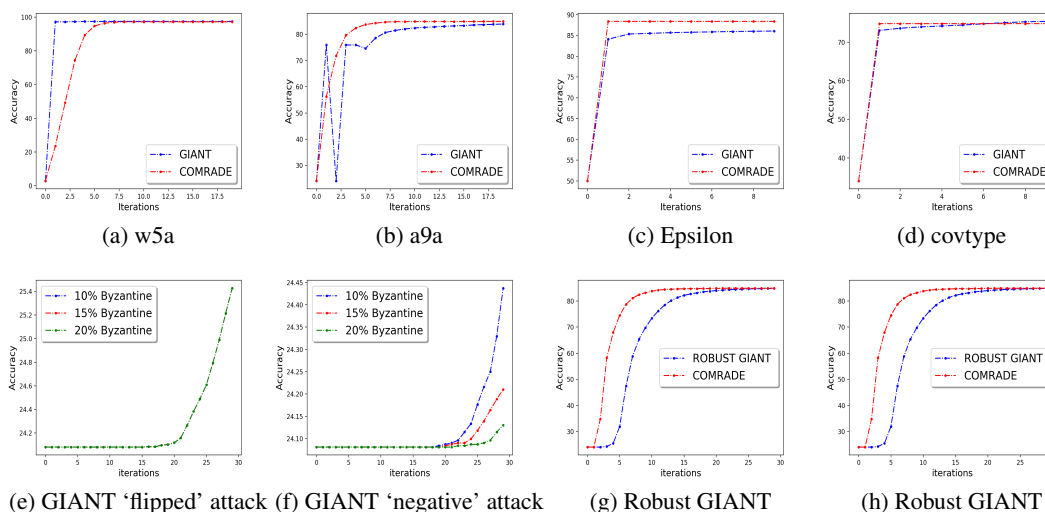

(a) w5a     (b) a9a     (c) Epsilon     (d) covtype

(e) GIANT 'flipped' attack (f) GIANT 'negative' attack     (g) Robust GIANT     (h) Robust GIANT

Figure 1: (First row) Comparison of training accuracy between COMRADE(Algorithm 1) and GIANT [31] with (a) w5a (b) a9a (c) Epsilon (d) Covtype dataset. (Second row) Training accuracy of (e) GIANT for 'flipped label' and (f) 'negative update' attack; and comparison of Robust GIANT and COMRADE with a9a dataset for (g) 'flipped label' and (h) 'negative update' attack.

second round of communication, the non-trimmed worker machines send the updates (product of local Hessian inverse and the local gradient) to the center machine. We compare COMRADE with 'Robust GIANT' in Figure 1((g),(h)) with $10\%$ Byzantine worker machines for 'a9a' dataset. It is evident plot that COMRADE performs better than the 'Robust GIANT'.

Next we show the accuracy of COMRADE with different numbers of Byzantine worker machines. Here we choose $c = 0.9$. We show the accuracy for 'negetive update ' attack in Figure 2(first row) and 'flipped label' attack in Figure 2 (second row). Furthermore, we show that COMRADE works even when $\rho$-approximate compressor is applied to the updates. In Figure 2(Third row) we plot the tranning accuracies. For compression we apply the scheme known as QSGD [1]. Further experiments can be found in the supplementary material.

# 7 Conclusion and Future Work

In this paper, we address the issue of communication efficiency and Byzantine robustness via second order optimization and norm based thresholding respectively for strongly convex loss. Extending our setting to handle weakly convex and non-convex loss is of immediate interest. We would also like to exploit local averaging with second order optimization. Moreover, an import aspect, privacy, is not addressed in this work. We keep this as our future research direction.

## Broader Impact

The advent of computationally-intensive machine learning (ML) models has changed the technology landscape in the past decade. The most powerful learning models are also the most expensive to train. For example, OpenAI's GPT-3 language model has 175 billion parameters and takes USD 12 million to train[1]! On top of that machine learning training has a costly environmental footprint: recent study shows that training a transformer with neural architecture search can have as much as five times $CO_2$ emission of a standard car in its lifetime[2]. While the really expensive models are relatively rare, training of moderately large ML models is now ubiquitous over the data science industry and elsewhere. Most of the training of machine learning model today is performed in distributed platforms

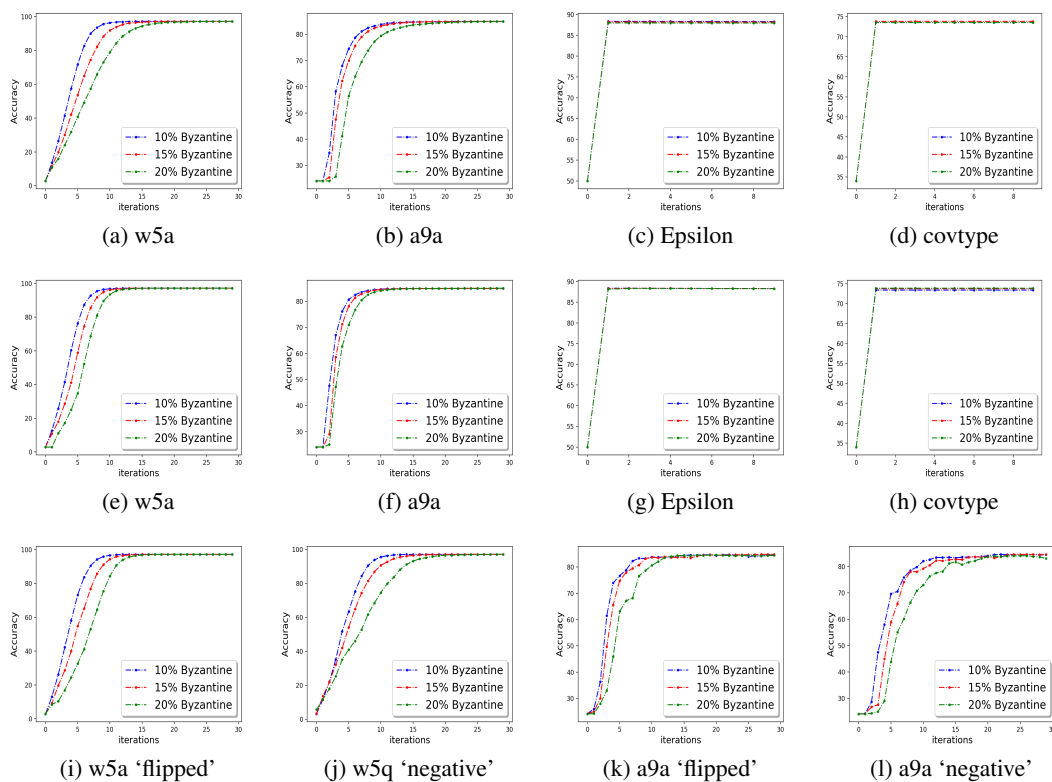

Figure 2: (First row) Accuracy of COMRADE with $10\%, 15\%, 20\%$ Byzantine workers with 'negative update' attack for (a). w5a (b). a9a (c). covtype (d). Epsilon. (Second row) COMRADE accuracy with $10\%, 15\%, 20\%$ Byzantine workers with 'flipped label' attack for (e) w5a (f) a9a (g) covtype (h) Epsilon. (Third row) Accuracy of COMRADE with $\rho$-approximate compressor (Section 5) with $10\%, 15\%, 20\%$ Byzantine workers; (i) 'flipped label' attack for w5a (j) 'negative update' attack for w5a. (k) 'flipped label' attack for a9a . (l) 'negative update' attack for a9a dataset.

(such as Amazon's EC2). Any savings in energy - in forms of computation or communication - in distributed optimization will have a large positive impact.

This paper seeks to speed up distributed optimization algorithms by minimizing inter-server communication and at the same time makes the optimization algorithms robust to adversarial failures. The protocols resulting from this paper are immediately implementable and can be adapted to any large scale distributed training of a machine learning model. Further, since our algorithms are robust to Byzantine failure, the training process becomes more reliable and fail-safe.

In addition to that, we think the theoretical content of this paper is instructive and some elements can be included in the coursework of a graduate class of distributed optimization, to exemplify the trade-off between some fundamental quantities in distributed optimization.

## Acknowledgment

We like to thank the reviewers for their insightful feedback. This work is supported by NSF grant 1642658.

## Footnotes

[1]`https://venturebeat.com/2020/06/01/ai-machine-learning-openai-gpt-3-size-isnt-everything/`

[2]MIT Tech. Review article dated 2019/06/06

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
