[Supplementary Material 1 · second.pdf]

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

# Distributed Newton Can Communicate Less and Resist Byzantine Workers
## Supplementary Material

## 8 Appendix A: Analysis of Section 3

**Matrix Sketching**

Here we briefly discuss the matrix sketching that is broadly used in the context of *randomized linear algebra*. For any matrix $\mathbf{A} \in \mathbb{R}^{n \times d}$ the sketched matrix $\mathbf{Z} \in \mathbb{R}^{s \times d}$ is defined as $\mathbf{S}^T \mathbf{A}$ where $\mathbf{S} \in \mathbb{R}^{n \times s}$ is the sketching matrix (typically $s < n$). Based on the scope and basis of the application, the sketched matrix is constructed by taking linear combination of the rows of matrix which is known as *random projection* or by sampling and scaling a subset of the rows of the matrix which is known as *random sampling*. The sketching is done to get a smaller representation of the original matrix to reduce computational cost.

Here we consider a uniform row sampling scheme. The matrix $\mathbf{Z}$ is formed by sampling and scaling rows of the matrix $\mathbf{A}$. Each row of the matrix $\mathbf{A}$ is sampled with probability $p = \frac{1}{n}$ and scaled by multiplying with $\frac{1}{\sqrt{sp}}$ .

$$\mathbb{P}\left(\mathbf{z}_i = \frac{\mathbf{a}_j}{\sqrt{sp}}\right) = p,$$

where $\mathbf{z}_i$ is the $i$-th row matrix $\mathbf{Z}$ and $\mathbf{a}_j$ is the $j$ th row of the matrix $\mathbf{A}$. Consequently the sketching matrix $\mathbf{S}$ has one non-zero entry in each column.

We define the matrix $\mathbf{A}_t^\top = [\mathbf{a}_1^\top, \ldots, \mathbf{a}_n^\top] \in \mathbb{R}^{d \times n}$ where $\mathbf{a}_j = \sqrt{\ell_j''(\mathbf{w}^\top \mathbf{x}_j)}\, \mathbf{x}_j$. So the exact Hessian in equation (2) is $\mathbf{H}_t = \frac{1}{n} \mathbf{A}_t^\top \mathbf{A}_t + \lambda \mathbf{I}$. Assume that $S_i$ is the set of features that are held by the $i$th worker machine. So the local Hessian is

$$\mathbf{H}_{i,t} = \frac{1}{s} \sum_{j \in S_i} \ell_j''(\mathbf{w}^\top \mathbf{x}_j) \mathbf{x}_j \mathbf{x}_j^\top + \lambda \mathbf{I} = \frac{1}{s} \mathbf{A}_{i,t}^\top \mathbf{A}_{i,t} + \lambda \mathbf{I},$$

where $\mathbf{A}_{i,t} \in \mathbb{R}^{s \times d}$ and the row of the matrix $\mathbf{A}_{i,t}$ is indexed by $S_i$. Also we define $\mathbf{B}_t = [\mathbf{b}_1, \ldots, \mathbf{b}_n] \in \mathbb{R}^{d \times n}$ where $\mathbf{b}_i = \ell_i'(\mathbf{w}^\top \mathbf{x}_i) \mathbf{x}_i$. So the exact gradient in equation (2) is $\mathbf{g}_t = \frac{1}{n} \mathbf{B}_t \mathbf{1} + \lambda \mathbf{w}_t$ and the local gradient is

$$\mathbf{g}_{i,t} = \frac{1}{s} \sum_{i \in S_i} \ell_j'(\mathbf{w}_t^\top \mathbf{x}_i) \mathbf{x}_i + \lambda \mathbf{w}_t = \frac{1}{s} \mathbf{B}_{i,t} \mathbf{1} + \lambda \mathbf{w}_t,$$

where $\mathbf{B}_{i,t}$ is the matrix with column indexed by $S_i$. If $\{\mathbf{S}_i\}_{i=1}^m$ are the sketching matrices then the local Hessian and gradient can be expressed as

$$\mathbf{H}_{i,t} = \mathbf{A}_t^\top \mathbf{S}_i \mathbf{S}_i^\top \mathbf{A}_t^\top + \lambda \mathbf{I} \qquad\qquad \mathbf{g}_{i,t} = \frac{1}{n} \mathbf{B} \mathbf{S}_i \mathbf{S}_i^\top \mathbf{1} + \lambda \mathbf{w}. \qquad (9)$$

With the help of sketching idea later we show that the local hessian and gradient are close to the exact hessian and gradient.

**The Quadratic function** For the purpose of analysis we define an auxiliary quadratic function

$$\phi(\mathbf{p}) = \frac{1}{2} \mathbf{p}^\top \mathbf{H}_t \mathbf{p} - \mathbf{g}_t^\top \mathbf{p} = \frac{1}{2} \mathbf{p}^\top (\mathbf{A}_t^\top \mathbf{A}_t + \lambda \mathbf{I}) \mathbf{p} - \mathbf{g}_t^\top \mathbf{p}. \qquad (10)$$

The optimal solution to the above function is

$$\mathbf{p}^* = \arg\min \phi(\mathbf{p}) = \mathbf{H}_t^{-1} \mathbf{g}_t = (\mathbf{A}_t^\top \mathbf{A}_t + \lambda \mathbf{I})^{-1} \mathbf{g}_t,$$

which is also the optimal direction of the global Newton update. In this work we consider the local and global (approximate ) Newton direction to be

$$\hat{\mathbf{p}}_{i,t} = (\mathbf{A}^\top \mathbf{S}_i \mathbf{S}_i^\top \mathbf{A} + \lambda \mathbf{I})^{-1} \mathbf{g}_{i,t}, \quad \hat{\mathbf{p}}_t = \frac{1}{m} \sum_{i=1}^m \hat{\mathbf{p}}_{i,t}.$$

respectively. And it can be easily verified that each local update $\hat{\mathbf{p}}_{i,t}$ is optimal solution to the following quadratic function

$$\hat{\phi}_{i,t}(p) = \frac{1}{2}\mathbf{p}^\top(\mathbf{A}^\top\mathbf{S}_i\mathbf{S}_i^\top\mathbf{A} + \lambda\mathbf{I})\mathbf{p} - \mathbf{g}_i^\top\mathbf{p}. \tag{11}$$

In our convergence analysis we show that value of the quadratic function in (10) with value $\hat{\mathbf{p}}_t$ is close to the optimal value.

**Singular Value Decomposition (SVD)** For any matrix $\mathbf{A} \in \mathbb{R}^{n \times d}$ with rank $r$, the singular value decomposition is defined as $\mathbf{A} = \mathbf{U}\boldsymbol{\Sigma}\mathbf{V}^\top$ where $\mathbf{U}, \mathbf{V}$ are $n \times r$ and $d \times r$ column orthogonal matrices respectively and $\boldsymbol{\Sigma}$ is a $r \times r$ diagonal matrix with diagonal entries $\{\sigma_1, \ldots \sigma_r\}$. If $\mathbf{A}$ is a symmetric positive semi-definite matrix then $\mathbf{U} = \mathbf{V}$.

## 8.1 Analysis

**Lemma 1** (McDiarmid's Inequality). *Let $X = X_1, \ldots, X_m$ be $m$ independent random variables taking values from some set $A$, and assume that $f : A^m \to \mathbb{R}$ satisfies the following condition (bounded differences ):*

$$\sup_{x_1, \ldots, x_m, \hat{x}_i} |f(x_i, \ldots, x_i, \ldots, x_m) - f(x_i, \ldots, \hat{x}_i, \ldots, x_m)| \le c_i,$$

*for all $i \in \{1, \ldots, m\}$. Then for any $\epsilon > 0$ we have*

$$P\left[f(X_1, \ldots, X_m) - \mathbb{E}[f(X_1, \ldots, X_m)] \ge \epsilon\right] \le \exp\left(-\frac{2\epsilon^2}{\sum_{i=1}^m c_i^2}\right).$$

The property described in the following Lemma 2 is a very useful result for uniform row sampling sketching matrix.

**Lemma 2** (Lemma 8 [30]). *Let $\eta, \delta \in (0, 1)$ be a fixed parameter and $r = rank(\mathbf{A}_t)$ and $\mathbf{U} \in \mathbb{R}^{n \times r}$ be the orthonormal bases of the matrix $\mathbf{A}_t$. Let $\{\mathbf{S}_i\}_{i=1}^m$ be sketching matrices and $\mathbf{S} = \frac{1}{\sqrt{m}}[\mathbf{S}_1, \ldots \mathbf{S}_m] \in \mathbb{R}^{n \times ms}$. With probability $1 - \delta$ the following holds*

$$\left\|\mathbf{U}^\top\mathbf{S}_i\mathbf{S}_i^\top\mathbf{U} - \mathbf{I}\right\|_2 \le \eta \quad \forall i \in [m] \quad and \quad \left\|\mathbf{U}^\top\mathbf{S}\mathbf{S}^\top\mathbf{U} - \mathbf{I}\right\|_2 \le \frac{\eta}{\sqrt{m}}.$$

**Lemma 3.** *Let $\mathbf{S} \in \mathbb{R}^{n \times s}$ be any uniform sampling sketching matrix, then for any matrix $\mathbf{B} = [\mathbf{b}_1, \ldots, \mathbf{b}_n] \in \mathbb{R}^{d \times n}$ with probability $1 - \delta$ for any $\delta > 0$ we have,*

$$\|\frac{1}{n}\mathbf{B}\mathbf{S}\mathbf{S}^\top\mathbf{1} - \frac{1}{n}\mathbf{B}\mathbf{1}\| \le (1 + \sqrt{2\ln(\frac{1}{\delta})})\sqrt{\frac{1}{s}}\max_i\|\mathbf{b}_i\|,$$

*where $\mathbf{1}$ is all ones vector.*

*Proof.* The vector $\mathbf{B}\mathbf{1}$ is the sum of column of the matrix $\mathbf{B}$ and $\mathbf{B}\mathbf{S}\mathbf{S}^\top\mathbf{1}$ is the sum of uniformly sampled and scaled column of the matrix $\mathbf{B}$ where the scaling factor is $\frac{1}{\sqrt{sp}}$ with $p = \frac{1}{n}$. If $(i_1, \ldots, i_s)$ is the set of sampled indices then $\mathbf{B}\mathbf{S}\mathbf{S}^\top\mathbf{1} = \sum_{k \in (i_1, \ldots, i_s)}\frac{1}{sp}\mathbf{b}_k$.

Define the function $f(i_1, \ldots, i_s) = \|\frac{1}{n}\mathbf{B}\mathbf{S}\mathbf{S}^\top\mathbf{1} - \frac{1}{n}\mathbf{B}\mathbf{1}\|$. Now consider a sampled set $(i_1, \ldots, i_{j'}, \ldots, i_s)$ with only one item (column) replaced then the bounded difference is

$$\Delta = |f(i_1, \ldots, i_j, \ldots, i_s) - f(i_1, \ldots, i_{j'}, \ldots, i_s)|$$
$$= |\frac{1}{n}\|\frac{1}{sp}\mathbf{b}_{i'_j} - \frac{1}{sp}\mathbf{b}_{i_j}\|| \le \frac{2}{s}\max_i\|\mathbf{b}_i\|.$$

Now we have the expectation

$$\mathbb{E}[\|\frac{1}{n}\mathbf{B}\mathbf{S}\mathbf{S}^\top\mathbf{1} - \frac{1}{n}\mathbf{B}\mathbf{1}\|^2] \le \frac{n}{sn^2}\sum_{i=1}^n\|\mathbf{b}_i\|^2 = \frac{1}{s}\max_i\|\mathbf{b}_i\|^2$$

$$\Rightarrow \mathbb{E}[\|\frac{1}{n}\mathbf{B}\mathbf{S}\mathbf{S}^\top\mathbf{1} - \frac{1}{n}\mathbf{B}\mathbf{1}\|] \le \sqrt{\frac{1}{s}}\max_i\|\mathbf{b}_i\|.$$

459  Using McDiarmid inequality (Lemma 1) we have

$$P[\left\|\frac{1}{n}\mathbf{B}\mathbf{S}\mathbf{S}^\top\mathbf{1} - \frac{1}{n}\mathbf{B}\mathbf{1}\right\| \geq \sqrt{\frac{1}{s}}\max_i\|\mathbf{b}_i\| + t] \leq \exp\left(-\frac{2t^2}{s\Delta^2}\right).$$

460  Equating the probability with $\delta$ we have

$$\exp(-\frac{2t^2}{s\Delta^2}) = \delta$$

$$\Rightarrow t = \Delta\sqrt{\frac{s}{2}\ln(\frac{1}{\delta})} = \max_i\|\mathbf{b}_i\|\sqrt{\frac{2}{s}\ln(\frac{1}{\delta})}.$$

461  Finally we have with probability $1 - \delta$

$$\|\frac{1}{n}\mathbf{B}\mathbf{S}\mathbf{S}^\top\mathbf{1} - \frac{1}{n}\mathbf{B}\mathbf{1}\| \leq (1 + \sqrt{2\ln(\frac{1}{\delta})})\sqrt{\frac{1}{s}}\max_i\|\mathbf{b}_i\|.$$

462  $\qquad\qquad\qquad\qquad\qquad\qquad\qquad\qquad\qquad\qquad\qquad\qquad\qquad\qquad\qquad\qquad\square$

463  **Remark 8.** *For $m$ sketching matrix $\{\mathbf{S}_i\}_{i=1}^m$, the bound in the Lemma 3 is*

$$\|\frac{1}{n}\mathbf{B}\mathbf{S}_i\mathbf{S}_i^\top\mathbf{1} - \frac{1}{n}\mathbf{B}\mathbf{1}\| \leq (1 + \sqrt{2\ln(\frac{m}{\delta})})\sqrt{\frac{1}{s}}\max_i\|\mathbf{b}_i\|,$$

464  *with probability $1 - \delta$ for any $\delta > 0$ for all $i \in \{1, 2, \ldots, m\}$. In the case that each worker machine*
465  *holds data based on the uniform sketching matrix the local gradient is close to the exact gradient.*
466  *Thus the local second order update acts as a good approximate to the exact Newton update.*

467  Now we consider the update rule of GIANT [30] where the update is done in two rounds in each
468  iteration. In the first round each worker machine computes and send the local gradient and the
469  center machine computes the exact gradient $\mathbf{g}_t$ in iteration $t$. Next the center machine broadcasts
470  the exact gradient and each worker machine computes the local Hessian and send $\tilde{\mathbf{p}}_{i,t} = (\mathbf{H}_{i,t})^{-1}\mathbf{g}_t$
471  to the center machine and the center machine computes the approximate Newton direction $\tilde{\mathbf{p}}_t = $
472  $\frac{1}{m}\sum_{i=1}^m \tilde{\mathbf{p}}_{i,t}$. Now based on this we restate the following lemma (Lemma 6 [30]).

473  **Lemma 4.** *Let $\{\mathbf{S}_i\}_{i=1}^m \in \mathbb{R}^{n \times s}$ be sketching matrices based on Lemma 2. Let $\phi_t$ be defined in (10)*
474  *and $\tilde{\mathbf{p}}_t$ be the update. It holds that*

$$\min_{\mathbf{p}} \phi_t(\mathbf{p}) \leq \phi_t(\tilde{\mathbf{p}}_t) \leq (1 - \zeta^2)\min_{\mathbf{p}} \phi_t(\mathbf{p}),$$

475  *where $\zeta = \nu(\frac{\eta}{\sqrt{m}} + \frac{\eta^2}{1-\eta})$ and $\nu = \frac{\sigma_{max}(\mathbf{A}^\top\mathbf{A})}{\sigma_{max}(\mathbf{A}^\top\mathbf{A})+n\lambda} \leq 1$.*

476  Now we prove similar guarantee for the update according to COMRADE in Algorithm 1.

477  **Lemma 5.** *Let $\{\mathbf{S}_i\}_{i=1}^m \in \mathbb{R}^{n \times s}$ be sketching matrices based on Lemma 2. Let $\phi_t$ be defined in (10)*
478  *and $\hat{\mathbf{p}}_t$ be defined in Algorithm 1($\beta = 0$)*

$$\min_{\mathbf{p}} \phi_t(\mathbf{p}) \leq \phi_t(\hat{\mathbf{p}}_t) \leq \epsilon^2 + (1 - \zeta^2)\min_{\mathbf{p}} \phi_t(\mathbf{p}),$$

479  *where $\epsilon = \frac{1}{1-\eta}\frac{1}{\sqrt{\sigma_{min}(\mathbf{H}_t)}}(1 + \sqrt{2\ln(\frac{m}{\delta})})\sqrt{\frac{1}{s}}\max_i\|\mathbf{b}_i\|$ and $\zeta = \nu(\frac{\eta}{\sqrt{m}} + \frac{\eta^2}{1-\eta})$ and $\nu = $*
480  *$\frac{\sigma_{max}(\mathbf{A}^\top\mathbf{A})}{\sigma_{max}(\mathbf{A}^\top\mathbf{A})+n\lambda}$.*

481  *Proof.* First consider the quadratic function (10)

$$\phi_t(\hat{\mathbf{p}}_t) - \phi_t(\mathbf{p}^*) = \frac{1}{2}\|\mathbf{H}_t^{\frac{1}{2}}(\hat{\mathbf{p}}_t - \mathbf{p}^*)\|^2$$

$$\leq \underbrace{(\|\mathbf{H}_t^{\frac{1}{2}}(\hat{\mathbf{p}}_t - \tilde{\mathbf{p}}_t)\|^2}_{Term1} + \underbrace{\|\mathbf{H}_t^{\frac{1}{2}}(\tilde{\mathbf{p}}_t - \mathbf{p}^*)\|)^2}_{Term2}, \qquad (12)$$

482 where $\tilde{\mathbf{p}}_t = \frac{1}{m}\sum_{i=1}^m (\mathbf{H}_{i,t})^{-1}\mathbf{g}_t$. First we bound the Term 2 of (12) using the quadratic function and
483 Lemma 4

$$\frac{1}{2}\left\|\mathbf{H}_t^{\frac{1}{2}}(\tilde{\mathbf{p}}_t - \mathbf{p}^*)\|\right)^2 \leq \zeta^2 \left\|\mathbf{H}_t^{\frac{1}{2}}\mathbf{p}^*\right\|^2 \quad \text{(Using Lemma 4 )}$$
$$= -\zeta^2\phi_t(\mathbf{p}^*). \tag{13}$$

484 The step in equation (13) is from the definition of the function $\phi_t$ and $\mathbf{p}^*$. It can be shown that

$$\phi_t(\mathbf{p}^*) = -\left\|\mathbf{H}_t^{\frac{1}{2}}\mathbf{p}^*\right\|^2.$$

485 Now we bound the Term 1 in (12). By Lemma 2, we have $(1-\eta)\mathbf{A}_t^\top\mathbf{A}_t \preceq \mathbf{A}_t^\top\mathbf{S}_i\mathbf{S}_i^\top\mathbf{A}_t \preceq$
486 $(1+\eta)\mathbf{A}_t^\top\mathbf{A}_t$. Following we have $(1-\eta)\mathbf{H}_t \preceq \mathbf{H}_{i,t} \preceq (1+\eta)\mathbf{H}_t$. Thus there exists matrix $\xi_i$
487 satisfying

$$\mathbf{H}_t^{\frac{1}{2}}\mathbf{H}_{i,t}^{-1}\mathbf{H}_t^{\frac{1}{2}} = \mathbf{I} + \xi_i \quad \text{and} \quad -\frac{\eta}{1+\eta} \preceq \xi_i \preceq \frac{\eta}{1-\eta},$$

488 So we have,

$$\left\|\mathbf{H}_t^{\frac{1}{2}}\mathbf{H}_{i,t}^{-1}\mathbf{H}_t^{\frac{1}{2}}\right\| \leq 1 + \frac{\eta}{1-\eta} = \frac{1}{1-\eta}. \tag{14}$$

489 Now we have

$$\left\|\mathbf{H}_t^{\frac{1}{2}}(\hat{\mathbf{p}}_t - \tilde{\mathbf{p}}_t)\right\| = \left\|\mathbf{H}_t^{\frac{1}{2}}\frac{1}{m}\sum_{i=1}^m (\hat{\mathbf{p}}_{i,t} - \tilde{\mathbf{p}}_{i,t})\right\|$$

$$\leq \frac{1}{m}\sum_{i=1}^m \left\|\mathbf{H}_t^{\frac{1}{2}}(\hat{\mathbf{p}}_{i,t} - \tilde{\mathbf{p}}_{i,t})\right\|$$

$$= \frac{1}{m}\sum_{i=1}^m \left\|\mathbf{H}_t^{\frac{1}{2}}\mathbf{H}_{i,t}^{-1}(\mathbf{g}_{i,t} - \mathbf{g}_t)\right\|$$

$$= \frac{1}{m}\sum_{i=1}^m \left\|\mathbf{H}_t^{\frac{1}{2}}\mathbf{H}_{i,t}^{-1}\mathbf{H}_t^{\frac{1}{2}}\mathbf{H}_t^{-\frac{1}{2}}(\mathbf{g}_{i,t} - \mathbf{g}_t)\right\|$$

$$\leq \frac{1}{m}\sum_{i=1}^m \left\|\mathbf{H}_t^{\frac{1}{2}}\mathbf{H}_{i,t}^{-1}\mathbf{H}_t^{\frac{1}{2}}\right\|\left\|\mathbf{H}_t^{-\frac{1}{2}}(\mathbf{g}_{i,t} - \mathbf{g}_t)\right\|$$

$$\leq \frac{1}{1-\eta}\frac{1}{m}\sum_{i=1}^m \left\|\mathbf{H}_t^{-\frac{1}{2}}(\mathbf{g}_{i,t} - \mathbf{g}_t)\right\| \quad (\text{ Using (14)})$$

$$\leq \frac{1}{1-\eta}\frac{1}{\sqrt{\sigma_{min}(\mathbf{H}_t)}}\frac{1}{m}\sum_{i=1}^m \|(\mathbf{g}_{i,t} - \mathbf{g}_t)\|. \tag{15}$$

490 Now we bound $\|(\mathbf{g}_{i,t} - \mathbf{g}_t)\|$ using Lemma 3,

$$\|(\mathbf{g}_{i,t} - \mathbf{g}_t)\| = \|\frac{1}{n}\mathbf{B}\mathbf{S}\mathbf{S}^\top\mathbf{1} - \frac{1}{n}\mathbf{B}\mathbf{1}\| \leq (1 + \sqrt{2\ln(\frac{m}{\delta})})\sqrt{\frac{1}{s}}\max_i \|\mathbf{b}_i\|.$$

491 Plugging it into equation (15) we get,

$$\left\|\mathbf{H}_t^{\frac{1}{2}}(\hat{\mathbf{p}}_t - \tilde{\mathbf{p}}_t)\right\| \leq \frac{1}{1-\eta}\frac{1}{\sqrt{\sigma_{min}(\mathbf{H}_t)}}\frac{1}{m}\sum_{i=1}^m \|(\mathbf{g}_{i,t} - \mathbf{g}_t)\|$$

$$\leq \frac{1}{1-\eta}\frac{1}{\sqrt{\sigma_{min}(\mathbf{H}_t)}}(1 + \sqrt{2\ln(\frac{m}{\delta})})\sqrt{\frac{1}{s}}\max_i \|\mathbf{b}_i\|. \tag{16}$$

492 Now collecting the terms of (16) and (13) and plugging them into (12) we have

$$\phi_t(\hat{\mathbf{p}}_t) - \phi_t(\mathbf{p}^*) \leq \epsilon^2 - \zeta^2\phi_t(\mathbf{p}^*)$$
$$\Rightarrow \phi_t(\hat{\mathbf{p}}_t) \leq \epsilon^2 + (1 - \zeta^2)\phi_t(\mathbf{p}^*),$$

493 where $\epsilon$ is as defined in (4).

494 $\qquad\qquad\qquad\qquad\qquad\qquad\qquad\qquad\qquad\qquad\qquad\qquad\qquad\qquad\qquad\qquad\qquad\square$

**Lemma 6.** *Let* $\zeta \in (0,1)$, $\epsilon$ *be any fixed parameter. And* $\hat{p}_t$ *satisfies* $\phi_t(\hat{\mathbf{p}}_t) \leq \epsilon^2 + (1 - \zeta^2) \min_{\mathbf{p}} \phi_t(\mathbf{p})$. *Under the Assumption 1(Hessian L-Lipschitz) and* $\boldsymbol{\Delta}_t = \mathbf{w}_t - \mathbf{w}^*$ *satisfies*

$$\boldsymbol{\Delta}_{t+1}^\top \mathbf{H}_t \boldsymbol{\Delta}_{t+1} \leq L \|\boldsymbol{\Delta}_{t+1}\| \|\boldsymbol{\Delta}_t\|^2 + \frac{\zeta^2}{1 - \zeta^2} \boldsymbol{\Delta}_t^\top \mathbf{H}_t \boldsymbol{\Delta}_t + 2\epsilon^2.$$

*Proof.* We have $\mathbf{w}_{t+1} = \mathbf{w}_t - \hat{\mathbf{p}}_t$, $\boldsymbol{\Delta}_t = \mathbf{w}_t - \mathbf{w}^*$ and $\boldsymbol{\Delta}_{t+1} = \mathbf{w}_{t+1} - \mathbf{w}^*$. Also $\hat{\mathbf{p}}_t = \mathbf{w}_t - \mathbf{w}_{t+1} = \boldsymbol{\Delta}_t - \boldsymbol{\Delta}_{t+1}$. From the definition of $\phi$ we have,

$$\phi_t(\hat{\mathbf{p}}_t) = \frac{1}{2}(\boldsymbol{\Delta}_t - \boldsymbol{\Delta}_{t+1})^\top \mathbf{H}_t (\boldsymbol{\Delta}_t - \boldsymbol{\Delta}_{t+1}) - (\boldsymbol{\Delta}_t - \boldsymbol{\Delta}_{t+1})) \, \mathbf{g}_t,$$

$$(1 - \zeta^2)\phi_t(\frac{1}{(1 - \zeta^2)}\boldsymbol{\Delta}_t) = \frac{1}{2(1 - \zeta^2)}\boldsymbol{\Delta}_t^\top \mathbf{H}_t \boldsymbol{\Delta}_t - \boldsymbol{\Delta}_t^\top \mathbf{g}_t.$$

From the above two equation we have

$$\phi_t(\hat{\mathbf{p}}_t) - (1 - \zeta^2)\phi_t(\frac{1}{(1 - \zeta^2)}\boldsymbol{\Delta}_t)$$

$$= \frac{1}{2}\boldsymbol{\Delta}_{t+1}^\top \mathbf{H}_t \boldsymbol{\Delta}_{t+1} - \frac{1}{2}\boldsymbol{\Delta}_t^\top \mathbf{H}_t \boldsymbol{\Delta}_{t+1} + \frac{1}{2}\boldsymbol{\Delta}_{t+1}^\top \mathbf{g}_t - \frac{\zeta^2}{2(1 - \zeta^2)}\boldsymbol{\Delta}_t^\top \mathbf{H}_t \boldsymbol{\Delta}_t.$$

From Lemma 5 the following holds

$$\phi_t(\hat{\mathbf{p}}_t) \leq \epsilon^2 + (1 - \zeta^2) \min_{\mathbf{p}} \phi_t(\mathbf{p})$$

$$\leq \epsilon^2 + (1 - \zeta^2)\phi_t(\frac{1}{(1 - \zeta^2)}\boldsymbol{\Delta}_t).$$

So we have

$$\frac{1}{2}\boldsymbol{\Delta}_{t+1}^\top \mathbf{H}_t \boldsymbol{\Delta}_{t+1} - \boldsymbol{\Delta}_t^\top \mathbf{H}_t \boldsymbol{\Delta}_{t+1} + \boldsymbol{\Delta}_{t+1}^\top \mathbf{g}_t - \frac{\zeta^2}{2(1 - \zeta^2)}\boldsymbol{\Delta}_t^\top \mathbf{H}_t \boldsymbol{\Delta}_t \leq \epsilon^2. \qquad (17)$$

Consider $\mathbf{g}_t = \mathbf{g}(\mathbf{w}_t)$

$$\mathbf{g}(\mathbf{w}_t) = \mathbf{g}(\mathbf{w}^*) + \left( \int_0^1 \nabla^2 f(\mathbf{w}^* + z(\mathbf{w}_t - \mathbf{w}^*))dz \right) (\mathbf{w}_t - \mathbf{w}^*)$$

$$= \left( \int_0^1 \nabla^2 f(\mathbf{w}^* + z(\mathbf{w}_t - \mathbf{w}^*))dz \right) \boldsymbol{\Delta}_t \quad (\text{as } \mathbf{g}(\mathbf{w}^*) = 0).$$

Now we bound the following

$$\|\mathbf{H}_t \boldsymbol{\Delta}_t - \mathbf{g}(\mathbf{w}_t)\| \leq \|\boldsymbol{\Delta}_t\| \left\| \int_0^1 [\nabla^2 f(\mathbf{w}_t) - \nabla^2 f(\mathbf{w}^* + z(\mathbf{w}_t - \mathbf{w}^*))]dz \right\|$$

$$\leq \|\boldsymbol{\Delta}_t\| \int_0^1 \left\| [\nabla^2 f(\mathbf{w}_t) - \nabla^2 f(\mathbf{w}^* + z(\mathbf{w}_t - \mathbf{w}^*))] \right\| dz \quad (\text{By Jensen's Inequality})$$

$$\leq \|\boldsymbol{\Delta}_t\| \int_0^1 (1 - z)L \|\mathbf{w}_t - \mathbf{w}^*\| dz \quad (\text{by } L\text{-Lipschitz assumption})$$

$$= \frac{L}{2} \|\boldsymbol{\Delta}_t\|^2.$$

Plugging it into (17) we have

$$\boldsymbol{\Delta}_{t+1}^\top \mathbf{H}_t \boldsymbol{\Delta}_{t+1} \leq 2\boldsymbol{\Delta}_{t+1}^\top (\mathbf{H}_t \boldsymbol{\Delta}_t - \mathbf{g}_t) + \frac{\zeta^2}{(1 - \zeta^2)}\boldsymbol{\Delta}_t^\top \mathbf{H}_t \boldsymbol{\Delta}_t + 2\epsilon^2$$

$$\leq 2 \|\boldsymbol{\Delta}_{t+1}\| \|\mathbf{H}_t \boldsymbol{\Delta}_t - \mathbf{g}_t\| + \frac{\zeta^2}{(1 - \zeta^2)}\boldsymbol{\Delta}_t^\top \mathbf{H}_t \boldsymbol{\Delta}_t + 2\epsilon^2$$

$$\leq L \|\boldsymbol{\Delta}_{t+1}\| \|\boldsymbol{\Delta}_t\|^2 + \frac{\zeta^2}{(1 - \zeta^2)}\boldsymbol{\Delta}_t^\top \mathbf{H}_t \boldsymbol{\Delta}_t + 2\epsilon^2.$$

$\square$

 **Proof of Theorem 1**

 *Proof.* From the Lemma 6 with probability $1 - \delta$

$$\boldsymbol{\Delta}_{t+1}^{\top}\mathbf{H}_t\boldsymbol{\Delta}_{t+1} \leq L\,\|\boldsymbol{\Delta}_{t+1}\|\,\|\boldsymbol{\Delta}_t\|^2 + \frac{\zeta^2}{(1-\zeta^2)}\boldsymbol{\Delta}_t^{\top}\mathbf{H}_t\boldsymbol{\Delta}_t + 2\epsilon^2$$

$$\leq L\|\boldsymbol{\Delta}_{t+1}\|\|\boldsymbol{\Delta}_t\|^2 + (\frac{\zeta^2}{1-\zeta^2}\sigma_{max}(\mathbf{H}_t))\|\boldsymbol{\Delta}_t\|^2 + 2\epsilon^2.$$

 So we have,

$$\|\boldsymbol{\Delta}_{t+1}\| \leq \max\{\sqrt{\frac{\sigma_{max}(\mathbf{H}_t)}{\sigma_{min}(\mathbf{H}_t)}(\frac{\zeta^2}{1-\zeta^2})}\|\boldsymbol{\Delta}_t\|, \frac{L}{\sigma_{min}(\mathbf{H}_t)}\|\boldsymbol{\Delta}_t\|^2\} + \frac{2\epsilon}{\sqrt{\sigma_{min}(\mathbf{H}_t)}}.$$

 $\square$

 # 9 Appendix B: Analysis of Section 4

 In this section we provide the theoretical analysis of the Byzantine robust method explained in
 Section 4 and prove the statistical guarantee. In any iteration $t$ the following holds

$$|\mathcal{U}_t| = |(\mathcal{U}_t \cap \mathcal{M}_t)| + |(\mathcal{U}_t \cap \mathcal{B}_t)|$$
$$|\mathcal{M}_t| = |(\mathcal{U}_t \cap \mathcal{M}_t)| + |(\mathcal{M}_t \cap \mathcal{T}_t)|.$$

 Combining both we have

$$|\mathcal{U}_t| = |\mathcal{M}_t| - |(\mathcal{M}_t \cap \mathcal{T}_t)| + |(\mathcal{U}_t \cap \mathcal{B}_t)|.$$

 **Lemma 7.** *Let* $\{\mathbf{S}_i\}_{i=1}^m \in \mathbb{R}^{n \times s}$ *be sketching matrices based on Lemma 2. Let* $\phi_t$ *be defined in* (10)
 *and* $\hat{\mathbf{p}}_t$ *be defined in Algorithm 1. It holds that*

$$\min_{\mathbf{p}} \phi_t(\mathbf{p}) \leq \phi_t(\hat{\mathbf{p}}_t) \leq \epsilon_{byz}^2 + (1 - \zeta_{byz}^2)\phi(\mathbf{p}^*),$$

 *where* $\epsilon_{byz}$ *and* $\zeta_{byz}$ *is defined in* (5) *and* (6) *respectively.*

 *Proof.* In the following analysis we omit the subscript '$t$'. From the definition of the quadratic
 function (10) we know that

$$\phi(\hat{\mathbf{p}}) - \phi(\mathbf{p}^*) = \frac{1}{2}\|\mathbf{H}^{\frac{1}{2}}(\hat{\mathbf{p}} - \mathbf{p}^*)\|^2.$$

 Now we consider

$$\frac{1}{2}\|\mathbf{H}^{\frac{1}{2}}(\hat{\mathbf{p}} - \mathbf{p}^*)\|^2 = \frac{1}{2}\|\mathbf{H}^{\frac{1}{2}}(\frac{1}{|\mathcal{U}|}\sum_{i \in \mathcal{U}}\hat{\mathbf{p}}_i - \mathbf{p}^*)\|^2$$

$$= \frac{1}{2}\|\mathbf{H}^{\frac{1}{2}}\frac{1}{|\mathcal{U}|}(\sum_{i \in \mathcal{M}}(\hat{\mathbf{p}}_i - \mathbf{p}^*) - \sum_{i \in (\mathcal{M} \cap \mathcal{T})}(\hat{\mathbf{p}}_i - \mathbf{p}^*) + \sum_{i \in (\mathcal{U} \cap \mathcal{B})}(\hat{\mathbf{p}}_i - \mathbf{p}^*))\|^2$$

$$\leq \underbrace{\|\mathbf{H}^{\frac{1}{2}}\frac{1}{|\mathcal{U}|}(\sum_{i \in \mathcal{M}}(\hat{\mathbf{p}}_i - \mathbf{p}^*)\|^2}_{Term1} + \underbrace{2\|\mathbf{H}^{\frac{1}{2}}\frac{1}{|\mathcal{U}|}\sum_{i \in (\mathcal{M} \cap \mathcal{T})}(\hat{\mathbf{p}}_i - \mathbf{p}^*)\|^2}_{Term2} + \underbrace{2\|\mathbf{H}^{\frac{1}{2}}\frac{1}{|\mathcal{U}|}\sum_{i \in (\mathcal{U} \cap \mathcal{B})}(\hat{\mathbf{p}}_i - \mathbf{p}^*))\|^2}_{Term3}.$$

 Now we bound each term separately and use the result of the Lemma 5 to bound each term.

$$Term1 = \|\mathbf{H}^{\frac{1}{2}}\frac{1}{|\mathcal{U}|}(\sum_{i \in \mathcal{M}}(\hat{\mathbf{p}}_i - \mathbf{p}^*)\|^2$$

$$= (\frac{1-\alpha}{1-\beta})^2\|\mathbf{H}^{\frac{1}{2}}\frac{1}{|\mathcal{M}|}(\sum_{i \in \mathcal{M}}(\hat{\mathbf{p}}_i - \mathbf{p}^*)\|^2$$

$$\leq (\frac{1-\alpha}{1-\beta})^2[\epsilon^2 + \zeta_{\mathcal{M}}^2\|\mathbf{H}^{\frac{1}{2}}\mathbf{p}^*\|^2],$$

521  where $\zeta_{\mathcal{M}} = \nu(\frac{\eta}{\sqrt{|\mathcal{M}|}} + \frac{\eta^2}{1-\eta}) = \nu(\frac{\eta}{\sqrt{(1-\alpha)m}} + \frac{\eta^2}{1-\eta})$.

522  Similarly the Term 2 can be bonded as it is a bound on good machines

$$Term2 = 2\|\mathbf{H}^{\frac{1}{2}} \frac{1}{|\mathcal{U}|} \sum_{i \in (\mathcal{M} \cap \mathcal{T})} (\hat{\mathbf{p}}_i - \mathbf{p}^*)\|^2$$

$$= 2(\frac{1-\alpha}{1-\beta})^2 \|\mathbf{H}^{\frac{1}{2}} \frac{1}{|\mathcal{M} \cap \mathcal{T}|} \sum_{i \in (\mathcal{M} \cap \mathcal{T})} (\hat{\mathbf{p}}_i - \mathbf{p}^*)\|^2$$

$$\leq 2(\frac{1-\alpha}{1-\beta})^2 [\epsilon^2 + \zeta_{\mathcal{M} \cap \mathcal{T}}^2 \|\mathbf{H}^{\frac{1}{2}} \mathbf{p}^*\|^2],$$

523  where $\zeta_{\mathcal{M} \cap \mathcal{T}} = \nu(\frac{\eta}{\sqrt{|\mathcal{M} \cap \mathcal{T}|}} + \frac{\eta^2}{1-\eta}) \leq \nu(\frac{\eta}{\sqrt{(1-\beta)m}} + \frac{\eta^2}{1-\eta})$.

524  For the Term 3 we know that $\beta > \alpha$ so all the untrimmed worker norm is bounded by a good machine
525  as at least one good machine gets trimmed.

$$Term3 = 2\|\mathbf{H}^{\frac{1}{2}} \frac{1}{|\mathcal{U}|} \sum_{i \in (\mathcal{U} \cap \mathcal{B})} (\hat{\mathbf{p}}_i - \mathbf{p}^*))\|^2$$

$$\leq 2\sigma_{max}(\mathbf{H})(\frac{|\mathcal{U} \cap \mathcal{B}|}{|\mathcal{U}|})^2 \|\frac{1}{|\mathcal{U} \cap \mathcal{B}|} \sum_{i \in (\mathcal{U} \cap \mathcal{B})} (\hat{\mathbf{p}}_i - \mathbf{p}^*))\|^2$$

$$\leq 2\sigma_{max}(\mathbf{H})(\frac{|\mathcal{U} \cap \mathcal{B}|}{|\mathcal{U}|})^2 \frac{1}{|\mathcal{U} \cap \mathcal{B}|} \sum_{i \in (\mathcal{U} \cap \mathcal{B})} \|(\hat{\mathbf{p}}_i - \mathbf{p}^*))\|^2$$

$$\leq 4\sigma_{max}(\mathbf{H})(\frac{|\mathcal{U} \cap \mathcal{B}|}{|\mathcal{U}|})^2 \frac{1}{|\mathcal{U} \cap \mathcal{B}|} \sum_{i \in (\mathcal{U} \cap \mathcal{B})} (\|\hat{\mathbf{p}}_i\|^2 + \|\mathbf{p}^*\|^2)$$

$$\leq 4\sigma_{max}(\mathbf{H})(\frac{|\mathcal{U} \cap \mathcal{B}|}{|\mathcal{U}|})^2 \max_{i \in \mathcal{M}}(\|\hat{\mathbf{p}}_i\|^2 + \|\mathbf{p}^*\|^2)$$

$$\leq 4\sigma_{max}(\mathbf{H})(\frac{|\mathcal{U} \cap \mathcal{B}|}{|\mathcal{U}|})^2 \max_{i \in \mathcal{M}}(\|\hat{\mathbf{p}}_i - \mathbf{p}^*\|^2 + 2\|\mathbf{p}^*\|^2)$$

$$\leq 4\kappa(\frac{|\mathcal{U} \cap \mathcal{B}|}{|\mathcal{U}|})^2 \max_{i \in \mathcal{M}}(\|\mathbf{H}^{\frac{1}{2}}(\hat{\mathbf{p}}_i - \mathbf{p}^*)\|^2 + 2\|\mathbf{H}^{\frac{1}{2}}\mathbf{p}^*\|^2)$$

$$\leq 4\kappa(\frac{|\mathcal{U} \cap \mathcal{B}|}{|\mathcal{U}|})^2 (\epsilon^2 + (2 + \zeta_1^2)\|\mathbf{H}^{\frac{1}{2}}\mathbf{p}^*\|^2)$$

$$\leq 4\kappa(\frac{\alpha}{1-\beta})^2 (\epsilon^2 + (2 + \zeta_1^2)\|\mathbf{H}^{\frac{1}{2}}\mathbf{p}^*\|^2),$$

526  where $\zeta_1 = \nu(\eta + \frac{\eta^2}{1-\eta}) = \frac{\nu}{1-\eta}$ and $\kappa = \frac{\sigma_{max}(\mathbf{H})}{\sigma_{min}(\mathbf{H})}$.

527  Combining all the bounds on Term1 , Term2 and Term3 we have

$$\frac{1}{2}\|\mathbf{H}^{\frac{1}{2}}(\hat{\mathbf{p}} - \mathbf{p}^*)\|^2 \leq \epsilon_{byz}^2 + \zeta_{byz}^2 \|\mathbf{H}^{\frac{1}{2}}\mathbf{p}^*\|^2,$$

528  where

$$\epsilon_{byz}^2 = \left(3\left(\frac{1-\alpha}{1-\beta}\right)^2 + 4\kappa\left(\frac{\alpha}{1-\beta}\right)^2\right)\epsilon^2,$$

$$\zeta_{byz}^2 = 2\left(\frac{1-\alpha}{1-\beta}\right)^2 \zeta_{\mathcal{M} \cap \mathcal{T}}^2 + \left(\frac{1-\alpha}{1-\beta}\right)^2 \zeta_{\mathcal{M}}^2 + 4\kappa\left(\frac{\alpha}{1-\beta}\right)^2 (2 + \zeta_1^2).$$

529  Finally we have

$$\phi(\hat{\mathbf{p}}) - \phi(\mathbf{p}^*) \leq \epsilon_{byz}^2 - \zeta_{byz}^2 \phi(\mathbf{p}^*)$$

$$\Rightarrow \phi(\hat{\mathbf{p}}) \leq \epsilon_{byz}^2 + (1 - \zeta_{byz}^2)\phi(\mathbf{p}^*).$$

530  $\square$

**Lemma 8.** *Let $\zeta_{byz} \in (0,1)$, $\epsilon_{byz}$ be any fixed parameter. And $\hat{p}_t$ satisfies $\phi_t(\hat{\mathbf{p}}_t) \le \epsilon_{byz}^2 + (1 - \zeta_{byz}^2) \min_{\mathbf{p}} \phi_t(\mathbf{p})$. Under the Assumption 1(Hessian L-Lipschitz) and $\boldsymbol{\Delta}_t = \mathbf{w}_t - \mathbf{w}^*$ satisfies*

$$\boldsymbol{\Delta}_{t+1}^T \mathbf{H}_t \boldsymbol{\Delta}_{t+1} \le L \|\boldsymbol{\Delta}_{t+1}\| \|\boldsymbol{\Delta}_t\|^2 + \frac{\zeta_{byz}^2}{1 - \zeta_{byz}^2} \boldsymbol{\Delta}_t^T \mathbf{H}_t \boldsymbol{\Delta}_t + 2\epsilon_{byz}^2.$$

*Proof.* We choose $\zeta = \zeta_{byz}$ and $\epsilon = \epsilon_{byz}$ from the Lemma 7 and follow the proof of Lemma 6 to obtain the desired bound. $\square$

**Proof of Theorem 2**

*Proof.* We get the desired bound by developing from the result of the Lemma 8 and following the proof of Theorem 1 $\square$

# 10 Appendix C:Analysis of Section 5

First we prove the following lemma that will be useful in our subsequent calculations. Consider that $\mathcal{Q}(\hat{\mathbf{p}}) = \frac{1}{|B|} \sum_{i \in B} \mathcal{Q}(\hat{\mathbf{p}}_i)$. And also we use the following notation $\zeta_B = \nu(\frac{\eta}{\sqrt{|B|}} + \frac{\eta^2}{1-\eta})$, $\nu = \frac{\sigma_{max}(\mathbf{A}^\top \mathbf{A})}{\sigma_{max}(\mathbf{A}^\top \mathbf{A}) + n\lambda} \le 1$.

**Lemma 9.** *If $\mathcal{Q}(\hat{\mathbf{p}}_i)$ is the local update direction and $\mathbf{p}^*$ is the optimal solution to the quadratic function $\phi$ then*

$$\left\| \mathbf{H}^{\frac{1}{2}}(\mathcal{Q}(\hat{\mathbf{p}}_i) - \mathbf{p}^*) \right\|^2 \le 1 + \kappa(1-\rho))\epsilon^2 + (\zeta_B^2 + \kappa(1-\rho)((1 + \zeta_1^2)) \left\| \mathbf{H}^{\frac{1}{2}} \mathbf{p}^* \right\|^2,$$

*where $\mathbf{H}$ is the exact Hessian and*

$$\epsilon_1 = \sqrt{(1 + \kappa(1-\rho))}\epsilon,$$
$$\zeta_{comp,B}^2 = (\zeta_B^2 + \kappa(1-\rho)((1 + \zeta_1^2)).$$

*$\epsilon$ is defined in equation (4) and*

*Proof.*

$$\left\| \mathbf{H}^{\frac{1}{2}}(\mathcal{Q}(\hat{\mathbf{p}}) - \mathbf{p}^*) \right\|^2 = \left\| \mathbf{H}^{\frac{1}{2}}(\mathcal{Q}(\hat{\mathbf{p}}) - \hat{\mathbf{p}} + \hat{\mathbf{p}} - \mathbf{p}^*) \right\|^2$$

$$\le 2 \left( \underbrace{\left\| \mathbf{H}^{\frac{1}{2}}(\mathcal{Q}(\hat{\mathbf{p}}) - \hat{\mathbf{p}}) \right\|^2}_{Term1} + \underbrace{\left\| \mathbf{H}^{\frac{1}{2}}(\hat{\mathbf{p}} - \mathbf{p}^*) \right\|^2}_{Term2} \right). \quad (18)$$

Following the proof of Lemma 5 we get

$$\left\| \mathbf{H}^{\frac{1}{2}}(\hat{\mathbf{p}}_i - \mathbf{p}^*) \right\|^2 \le \epsilon^2 + \zeta_1 \left\| \mathbf{H}^{\frac{1}{2}} \mathbf{p}^* \right\|^2, \quad (19)$$

where $\epsilon$ is as defined in (4).Now we consider the term

$$\left\| \mathbf{H}^{\frac{1}{2}}(\mathcal{Q}(\hat{\mathbf{p}}_i) - \hat{\mathbf{p}}_i) \right\|^2 \le \sigma_{max}(\mathbf{H})(1-\rho) \|\hat{\mathbf{p}}_i\|^2$$

$$\le \sigma_{max}(\mathbf{H})(1-\rho) \left( \|\hat{\mathbf{p}}_i - \mathbf{p}^*\|^2 + \|\mathbf{p}^*\|^2 \right)$$

$$\le \frac{\sigma_{max}}{\sigma_{min}}(1-\rho) \left( \left\| \mathbf{H}^{\frac{1}{2}}(\hat{\mathbf{p}}_i - \mathbf{p}^*) \right\|^2 + \left\| \mathbf{H}^{\frac{1}{2}} \mathbf{p}^* \right\|^2 \right)$$

$$= \kappa(1-\rho) \left( \left\| \mathbf{H}^{\frac{1}{2}}(\hat{\mathbf{p}}_i - \mathbf{p}^*) \right\|^2 + \left\| \mathbf{H}^{\frac{1}{2}} \mathbf{p}^* \right\|^2 \right)$$

$$\le \kappa(1-\rho) \left( \epsilon^2 + (1 + \zeta_1^2) \left\| \mathbf{H}^{\frac{1}{2}} \mathbf{p}^* \right\|^2 \right) \quad \text{Using (19).}$$

548     Now we use the above calculation and bound Term1

$$\left\|\mathbf{H}^{\frac{1}{2}}(\mathcal{Q}(\hat{\mathbf{p}}) - \hat{\mathbf{p}})\right\|^2 \le \frac{1}{|B|} \sum_{i\in B} \left\|\mathbf{H}^{\frac{1}{2}}(\mathcal{Q}(\hat{\mathbf{p}}_i) - \hat{\mathbf{p}}_i)\right\|^2$$

$$\le \kappa(1-\rho)\left(\epsilon^2 + (1+\zeta_1^2)\left\|\mathbf{H}^{\frac{1}{2}}\mathbf{p}^*\right\|^2\right). \tag{20}$$

549     We can bound the Term2 directly using the proof of Lemma 5

$$\left\|\mathbf{H}^{\frac{1}{2}}(\hat{\mathbf{p}} - \mathbf{p}^*)\right\|^2 \le \epsilon^2 + \zeta_B^2 \left\|\mathbf{H}^{\frac{1}{2}}\mathbf{p}^*\right\|^2. \tag{21}$$

550     Now we use (20) and (21) and plug them in (18)

$$\left\|\mathbf{H}^{\frac{1}{2}}(\mathcal{Q}(\hat{\mathbf{p}}) - \mathbf{p}^*)\right\|^2 \le (1 + \kappa(1-\rho))\epsilon^2 + (\zeta_B^2 + \kappa(1-\rho)((1+\zeta_1^2))\left\|\mathbf{H}^{\frac{1}{2}}\mathbf{p}^*\right\|^2.$$

551     Now we define

$$\epsilon_1 = \sqrt{(1+\kappa(1-\rho))}\epsilon$$
$$\zeta_{comp,B}^2 = (\zeta_B^2 + \kappa(1-\rho)((1+\zeta_1^2)).$$

552                                                            □

553     Now we have the robust update in iteration $t$ to be $\mathcal{Q}(\hat{\mathbf{p}}) = \frac{1}{|\mathcal{U}_t|}\sum_{i\in\mathcal{U}_t} \mathcal{Q}(\hat{p}_{i,t})$.

554 **Lemma 10.** *Let $\{\mathbf{S}_i\}_{i=1}^m \in \mathbb{R}^{n\times s}$ be sketching matrices based on Lemma 2. Let $\phi_t$ be defined in*
555 *(10) and $\mathcal{Q}(\hat{\mathbf{p}}_t)$ be the update with $\mathcal{Q}$ being $\rho$-approximate compressor. It holds that*

$$\min_{\mathbf{p}} \phi_t(\mathbf{p}) \le \phi_t(\mathcal{Q}(\hat{\mathbf{p}}_t)) \le \epsilon_{comp,byz}^2 + (1 - \zeta_{comp,byz}^2)\phi_t(\mathbf{p}^*),$$

556 *where $\epsilon_{comp,byz}$ and $\zeta_{comp,byz}^2$ is as defined in (7) and (8) respectively.*

557 *Proof.* In the following analysis we omit the subscript '$t$'. From the definition of the quadratic
558 function (10) we know that

$$\phi(\mathcal{Q}(\hat{\mathbf{p}})) - \phi(\mathbf{p}^*) = \frac{1}{2}\|\mathbf{H}^{\frac{1}{2}}(\mathcal{Q}(\hat{\mathbf{p}}) - \mathbf{p}^*)\|^2.$$

559 Now we consider
$$\frac{1}{2}\|\mathbf{H}^{\frac{1}{2}}(\mathcal{Q}(\hat{\mathbf{p}}) - \mathbf{p}^*)\|^2 = \frac{1}{2}\|\mathbf{H}^{\frac{1}{2}}(\frac{1}{|\mathcal{U}|}\sum_{i\in\mathcal{U}} \mathcal{Q}(\hat{\mathbf{p}}_i) - \mathbf{p}^*)\|^2$$

$$= \frac{1}{2}\|\mathbf{H}^{\frac{1}{2}}\frac{1}{|\mathcal{U}|}(\sum_{i\in\mathcal{M}}(\mathcal{Q}(\hat{\mathbf{p}}_i) - \mathbf{p}^*) - \sum_{i\in(\mathcal{M}\cap\mathcal{T})}(\mathcal{Q}(\hat{\mathbf{p}}_i) - \mathbf{p}^*) + \sum_{i\in(\mathcal{U}\cap\mathcal{B})}(\mathcal{Q}(\hat{\mathbf{p}}_i) - \mathbf{p}^*))\|^2$$

$$\le \underbrace{\|\mathbf{H}^{\frac{1}{2}}\frac{1}{|\mathcal{U}|}(\sum_{i\in\mathcal{M}}(\mathcal{Q}(\hat{\mathbf{p}}_i) - \mathbf{p}^*)\|^2}_{Term1} + \underbrace{2\|\mathbf{H}^{\frac{1}{2}}\frac{1}{|\mathcal{U}|}\sum_{i\in(\mathcal{M}\cap\mathcal{T})}(\mathcal{Q}(\hat{\mathbf{p}}_i) - \mathbf{p}^*)\|^2}_{Term2}$$

$$+ \underbrace{2\|\mathbf{H}^{\frac{1}{2}}\frac{1}{|\mathcal{U}|}\sum_{i\in(\mathcal{U}\cap\mathcal{B})}(\mathcal{Q}(\hat{\mathbf{p}}_i) - \mathbf{p}^*))\|^2}_{Term3}.$$

560     Now we bound each term separately and use the Lemma 9

$$Term1 = \|\mathbf{H}^{\frac{1}{2}}\frac{1}{|\mathcal{U}|}(\sum_{i\in\mathcal{M}}(\mathcal{Q}(\hat{\mathbf{p}}_i) - \mathbf{p}^*)\|^2$$

$$= (\frac{1-\alpha}{1-\beta})^2\|\mathbf{H}^{\frac{1}{2}}\frac{1}{|\mathcal{M}|}(\sum_{i\in\mathcal{M}}(\mathcal{Q}(\hat{\mathbf{p}}_i) - \mathbf{p}^*)\|^2$$

$$\le (\frac{1-\alpha}{1-\beta})^2[\epsilon_1^2 + \zeta_{comp,\mathcal{M}}^2\|\mathbf{H}^{\frac{1}{2}}\mathbf{p}^*\|^2],$$

561 where $\zeta_{comp,\mathcal{M}}^2 = (\zeta_{\mathcal{M}}^2 + \kappa(1-\rho))((1+\zeta_1^2)$. Similarly the Term 2 can be bonded as it is a bound on
562 good machines

$$Term2 = 2\|\mathbf{H}^{\frac{1}{2}}\frac{1}{|\mathcal{U}|}\sum_{i\in(\mathcal{M}\cap\mathcal{T})}(\mathcal{Q}(\hat{\mathbf{p}}_i)-\mathbf{p}^*)\|^2$$

$$= 2(\frac{1-\alpha}{1-\beta})^2\|\mathbf{H}^{\frac{1}{2}}\frac{1}{|\mathcal{M}\cap\mathcal{T}|}\sum_{i\in(\mathcal{M}\cap\mathcal{T})}(\mathcal{Q}(\hat{\mathbf{p}}_i)-\mathbf{p}^*)\|^2$$

$$\leq 2(\frac{1-\alpha}{1-\beta})^2[\epsilon_1^2 + \zeta_{comp,\mathcal{M}\cap\mathcal{T}}^2\|\mathbf{H}^{\frac{1}{2}}\mathbf{p}^*\|^2].$$

563 For the Term 3 we know that $\beta > \alpha$ so all the untrimmed worker norm is bounded by a good machine
564 as at least one good machine gets trimmed.

$$Term3 = 2\|\mathbf{H}^{\frac{1}{2}}\frac{1}{|\mathcal{U}|}\sum_{i\in(\mathcal{U}\cap\mathcal{B})}(\mathcal{Q}(\hat{\mathbf{p}}_i)-\mathbf{p}^*))\|^2$$

$$\leq 2\sigma_{max}(\mathbf{H})(\frac{|\mathcal{U}\cap\mathcal{B}|}{|\mathcal{U}|})^2\|\frac{1}{|\mathcal{U}\cap\mathcal{B}|}\sum_{i\in(\mathcal{U}\cap\mathcal{B})}(\mathcal{Q}(\hat{\mathbf{p}}_i)-\mathbf{p}^*))\|^2$$

$$\leq 2\sigma_{max}(\mathbf{H})(\frac{|\mathcal{U}\cap\mathcal{B}|}{|\mathcal{U}|})^2\frac{1}{|\mathcal{U}\cap\mathcal{B}|}\sum_{i\in(\mathcal{U}\cap\mathcal{B})}\|(\mathcal{Q}(\hat{\mathbf{p}}_i)-\mathbf{p}^*))\|^2$$

$$\leq 4\sigma_{max}(\mathbf{H})(\frac{|\mathcal{U}\cap\mathcal{B}|}{|\mathcal{U}|})^2\frac{1}{|\mathcal{U}\cap\mathcal{B}|}\sum_{i\in(\mathcal{U}\cap\mathcal{B})}(\|\mathcal{Q}(\hat{\mathbf{p}}_i)\|^2 + \|\mathbf{p}^*\|^2)$$

$$\leq 4\sigma_{max}(\mathbf{H})(\frac{|\mathcal{U}\cap\mathcal{B}|}{|\mathcal{U}|})^2\max_{i\in\mathcal{M}}(\|\mathcal{Q}(\hat{\mathbf{p}}_i)\|^2 + \|\mathbf{p}^*\|^2)$$

$$\leq 4\sigma_{max}(\mathbf{H})(\frac{|\mathcal{U}\cap\mathcal{B}|}{|\mathcal{U}|})^2\max_{i\in\mathcal{M}}(\|\mathcal{Q}(\hat{\mathbf{p}}_i)-\mathbf{p}^*\|^2 + 2\|\mathbf{p}^*\|^2)$$

$$\leq 4\kappa(\frac{|\mathcal{U}\cap\mathcal{B}|}{|\mathcal{U}|})^2\max_{i\in\mathcal{M}}(\|\mathbf{H}^{\frac{1}{2}}(\mathcal{Q}(\hat{\mathbf{p}}_i)-\mathbf{p}^*)\|^2 + 2\|\mathbf{H}^{\frac{1}{2}}\mathbf{p}^*\|^2)$$

$$\leq 4\kappa(\frac{|\mathcal{U}\cap\mathcal{B}|}{|\mathcal{U}|})^2(\epsilon_1^2 + (2+\zeta_1^2)\|\mathbf{H}^{\frac{1}{2}}\mathbf{p}^*\|^2)$$

$$\leq 4\kappa(\frac{\alpha}{1-\beta})^2(\epsilon_1^2 + (2+\zeta_1^2)\|\mathbf{H}^{\frac{1}{2}}\mathbf{p}^*\|^2).$$

565 Combining all the bounds on Term1 , Term2 and Term3 we have

$$\frac{1}{2}\|\mathbf{H}^{\frac{1}{2}}(\hat{\mathbf{p}}-\mathbf{p}^*)\|^2 \leq \epsilon_{byz}^2 + \zeta_{byz}^2\|\mathbf{H}^{\frac{1}{2}}\mathbf{p}^*\|^2,$$

566 where

$$\epsilon_{comp,byz}^2 = \left(3\left(\frac{1-\alpha}{1-\beta}\right)^2 + 4\kappa\left(\frac{\alpha}{1-\beta}\right)^2\right)\epsilon_1^2$$

$$\zeta_{comp,byz}^2 = 2\left(\frac{1-\alpha}{1-\beta}\right)^2\zeta_{comp,\mathcal{M}\cap\mathcal{T}}^2 + \left(\frac{1-\alpha}{1-\beta}\right)^2\zeta_{comp,\mathcal{M}}^2 + 4\kappa\left(\frac{\alpha}{1-\beta}\right)^2(2+\zeta_{comp,1}^2).$$

567 Finally we have

$$\phi(\hat{\mathbf{p}}) - \phi(\mathbf{p}^*) \leq \epsilon_{comp,byz}^2 - \zeta_{comp,byz}^2\phi(\mathbf{p}^*)$$

$$\Rightarrow \phi(\hat{\mathbf{p}}) \leq \epsilon_{comp,byz}^2 + (1-\zeta_{comp,byz}^2)\phi(\mathbf{p}^*).$$

568 $\square$

569 **Lemma 11.** *Let* $\zeta_{comp,byz} \in (0,1), \epsilon_{comp,byz}$ *be any fixed parameter. And* $\mathcal{Q}(\hat{p}_t)$ *satisfies*
570 $\phi_t(\mathcal{Q}(\hat{p}_t)) \leq \epsilon_{byz}^2 + (1-\zeta_{byz}^2)\min_{\mathbf{p}}\phi_t(\mathbf{p})$. *Under the Assumption 1(Hessian L-Lipschitz) and*
571 $\mathbf{\Delta}_t = \mathbf{w}_t - \mathbf{w}^*$ *satisfies*

$$\mathbf{\Delta}_{t+1}^T\mathbf{H}_t\mathbf{\Delta}_{t+1} \leq L\|\mathbf{\Delta}_{t+1}\|\|\mathbf{\Delta}_t\|^2 + \frac{\zeta_{comp,byz}^2}{1-\zeta_{comp,byz}^2}\mathbf{\Delta}_t^T\mathbf{H}_t\mathbf{\Delta}_t + 2\epsilon_{comp,byz}^2.$$

*Proof.* We choose $\zeta = \zeta_{comp,byz}$ and $\epsilon = \epsilon_{comp,byz}$ from the Lemma 10 and follow the proof of Lemma 6 to obtain the desired bound. $\square$

**Proof of Theorem 3**

*Proof.* We get the desired bound by developing from the result of the Lemma 11 and following the proof of Theorem 1 $\square$

## 11  Additional Experiment

In addition to the experimental results in Section 6, we provide some more experiments supporting the robustness of the COMRADE in two different types of attacks : 1. 'Gaussian attack': where the Byzantine workers add Gaussian Noise $(\mathcal{N}(\mu, \sigma^2))$ to the update and 2. 'random label attack': where the Byzantine worker machines learns based on random labels instead of proper labels.

(a) w5a 'Gauss'       (b) a9a 'Gauss'       (c) w5a 'random'       (d) a9a 'random'

(e) w5a 'Gauss'       (f) a9a 'Gauss'       (g) w5a 'random'       (h) a9a 'random'

Figure 3: (First row) Accuracy of COMRADE with $10\%, 15\%, 20\%$ Byzantine workers with 'Gaussian ' attack for (a). w5a (b). a9a and 'random label' attack for (c). w5a (d).a9a. (Second row) Accuracy of COMRADE with $\rho$-approximate compressor (Section 5) with $10\%, 15\%, 20\%$ Byzantine workers with 'Gaussian ' attack for (a). w5a (b). a9a and 'random label' attack for (c). w5a (d).a9a.

[Supplementary Material 2 · second_supp.pdf]

# Distributed Newton Can Communicate Less and Resist Byzantine Workers
## Supplementary Material

## 8   Appendix A: Analysis of Section 3

**Matrix Sketching**
Here we briefly discuss the matrix sketching that is broadly used in the context of *randomized linear algebra*. For any matrix $\mathbf{A} \in \mathbb{R}^{n \times d}$ the sketched matrix $\mathbf{Z} \in \mathbb{R}^{s \times d}$ is defined as $\mathbf{S}^T \mathbf{A}$ where $\mathbf{S} \in \mathbb{R}^{n \times s}$ is the sketching matrix (typically $s < n$). Based on the scope and basis of the application, the sketched matrix is constructed by taking linear combination of the rows of matrix which is known as *random projection* or by sampling and scaling a subset of the rows of the matrix which is known as *random sampling*. The sketching is done to get a smaller representation of the original matrix to reduce computational cost.

Here we consider a uniform row sampling scheme. The matrix $\mathbf{Z}$ is formed by sampling and scaling rows of the matrix $\mathbf{A}$. Each row of the matrix $\mathbf{A}$ is sampled with probability $p = \frac{1}{n}$ and scaled by multiplying with $\frac{1}{\sqrt{sp}}$ .

$$\mathbb{P}\left(\mathbf{z}_i = \frac{\mathbf{a}_j}{\sqrt{sp}}\right) = p,$$

where $\mathbf{z}_i$ is the $i$-th row matrix $\mathbf{Z}$ and $\mathbf{a}_j$ is the $j$ th row of the matrix $\mathbf{A}$. Consequently the sketching matrix $\mathbf{S}$ has one non-zero entry in each column.

We define the matrix $\mathbf{A}_t^\top = [\mathbf{a}_1^\top, \ldots, \mathbf{a}_n^\top] \in \mathbb{R}^{d \times n}$ where $\mathbf{a}_j = \sqrt{\ell_j''(\mathbf{w}^\top \mathbf{x}_j)}\, \mathbf{x}_j$. So the exact Hessian in equation (2) is $\mathbf{H}_t = \frac{1}{n}\mathbf{A}_t^\top \mathbf{A}_t + \lambda \mathbf{I}$. Assume that $S_i$ is the set of features that are held by the $i$th worker machine. So the local Hessian is

$$\mathbf{H}_{i,t} = \frac{1}{s}\sum_{j \in S_i} \ell_j''(\mathbf{w}^\top \mathbf{x}_j)\mathbf{x}_j\mathbf{x}_j^\top + \lambda \mathbf{I} = \frac{1}{s}\mathbf{A}_{i,t}^\top \mathbf{A}_{i,t} + \lambda \mathbf{I},$$

where $\mathbf{A}_{i,t} \in \mathbb{R}^{s \times d}$ and the row of the matrix $\mathbf{A}_{i,t}$ is indexed by $S_i$. Also we define $\mathbf{B}_t = [\mathbf{b}_1, \ldots, \mathbf{b}_n] \in \mathbb{R}^{d \times n}$ where $\mathbf{b}_i = \ell_i'(\mathbf{w}^\top \mathbf{x}_i)\mathbf{x}_i$. So the exact gradient in equation (2) is $\mathbf{g}_t = \frac{1}{n}\mathbf{B}_t \mathbf{1} + \lambda \mathbf{w}_t$ and the local gradient is

$$\mathbf{g}_{i,t} = \frac{1}{s}\sum_{i \in S_i} \ell_j'(\mathbf{w}_t^\top \mathbf{x}_i)\mathbf{x}_i + \lambda \mathbf{w}_t = \frac{1}{s}\mathbf{B}_{i,t}\mathbf{1} + \lambda \mathbf{w}_t,$$

where $\mathbf{B}_{i,t}$ is the matrix with column indexed by $S_i$. If $\{\mathbf{S}_i\}_{i=1}^m$ are the sketching matrices then the local Hessian and gradient can be expressed as

$$\mathbf{H}_{i,t} = \mathbf{A}_t^\top \mathbf{S}_i \mathbf{S}_i^\top \mathbf{A}_t^\top + \lambda \mathbf{I} \qquad\qquad \mathbf{g}_{i,t} = \frac{1}{n}\mathbf{B}\mathbf{S}_i \mathbf{S}_i^\top \mathbf{1} + \lambda \mathbf{w}. \qquad (9)$$

With the help of sketching idea later we show that the local hessian and gradient are close to the exact hessian and gradient.

**The Quadratic function**  For the purpose of analysis we define an auxiliary quadratic function

$$\phi(\mathbf{p}) = \frac{1}{2}\mathbf{p}^\top \mathbf{H}_t \mathbf{p} - \mathbf{g}_t^\top \mathbf{p} = \frac{1}{2}\mathbf{p}^\top (\mathbf{A}_t^\top \mathbf{A}_t + \lambda \mathbf{I})\mathbf{p} - \mathbf{g}_t^\top \mathbf{p}. \qquad (10)$$

The optimal solution to the above function is

$$\mathbf{p}^* = \arg\min \phi(\mathbf{p}) = \mathbf{H}_t^{-1}\mathbf{g}_t = (\mathbf{A}_t^\top \mathbf{A}_t + \lambda \mathbf{I})^{-1}\mathbf{g}_t,$$

which is also the optimal direction of the global Newton update. In this work we consider the local and global (approximate ) Newton direction to be

$$\hat{\mathbf{p}}_{i,t} = (\mathbf{A}^\top \mathbf{S}_i \mathbf{S}_i^\top \mathbf{A} + \lambda \mathbf{I})^{-1}\mathbf{g}_{i,t}, \quad \hat{\mathbf{p}}_t = \frac{1}{m}\sum_{i=1}^m \hat{\mathbf{p}}_{i,t}.$$

respectively. And it can be easily verified that each local update $\hat{\mathbf{p}}_{i,t}$ is optimal solution to the following quadratic function

$$\hat{\phi}_{i,t}(p) = \frac{1}{2}\mathbf{p}^\top(\mathbf{A}^\top\mathbf{S}_i\mathbf{S}_i^\top\mathbf{A} + \lambda\mathbf{I})\mathbf{p} - \mathbf{g}_i^\top\mathbf{p}. \tag{11}$$

In our convergence analysis we show that value of the quadratic function in (10) with value $\hat{\mathbf{p}}_t$ is close to the optimal value.

**Singular Value Decomposition (SVD)** For any matrix $\mathbf{A} \in \mathbb{R}^{n \times d}$ with rank $r$, the singular value decomposition is defined as $\mathbf{A} = \mathbf{U}\mathbf{\Sigma}\mathbf{V}^\top$ where $\mathbf{U}, \mathbf{V}$ are $n \times r$ and $d \times r$ column orthogonal matrices respectively and $\mathbf{\Sigma}$ is a $r \times r$ diagonal matrix with diagonal entries $\{\sigma_1, \ldots \sigma_r\}$. If $\mathbf{A}$ is a symmetric positive semi-definite matrix then $\mathbf{U} = \mathbf{V}$.

## 8.1 Analysis

**Lemma 1** (McDiarmid's Inequality). *Let $X = X_1, \ldots, X_m$ be $m$ independent random variables taking values from some set $A$, and assume that $f : A^m \to \mathbb{R}$ satisfies the following condition (bounded differences ):*

$$\sup_{x_1,\ldots,x_m,\hat{x}_i} |f(x_i, \ldots, x_i, \ldots, x_m) - f(x_i, \ldots, \hat{x}_i, \ldots, x_m)| \le c_i,$$

*for all $i \in \{1, \ldots, m\}$. Then for any $\epsilon > 0$ we have*

$$P\left[f(X_1, \ldots, X_m) - \mathbb{E}[f(X_1, \ldots, X_m)] \ge \epsilon\right] \le \exp\left(-\frac{2\epsilon^2}{\sum_{i=1}^m c_i^2}\right).$$

The property described in the following Lemma 2 is a very useful result for uniform row sampling sketching matrix.

**Lemma 2** (Lemma 8 [30]). *Let $\eta, \delta \in (0, 1)$ be a fixed parameter and $r = rank(\mathbf{A}_t)$ and $\mathbf{U} \in \mathbb{R}^{n \times r}$ be the orthonormal bases of the matrix $\mathbf{A}_t$. Let $\{\mathbf{S}_i\}_{i=1}^m$ be sketching matrices and $\mathbf{S} = \frac{1}{\sqrt{m}}[\mathbf{S}_1, \ldots \mathbf{S}_m] \in \mathbb{R}^{n \times ms}$. With probability $1 - \delta$ the following holds*

$$\left\|\mathbf{U}^\top\mathbf{S}_i\mathbf{S}_i^\top\mathbf{U} - \mathbf{I}\right\|_2 \le \eta \quad \forall i \in [m] \quad and \quad \left\|\mathbf{U}^\top\mathbf{S}\mathbf{S}^\top\mathbf{U} - \mathbf{I}\right\|_2 \le \frac{\eta}{\sqrt{m}}.$$

**Lemma 3.** *Let $\mathbf{S} \in \mathbb{R}^{n \times s}$ be any uniform sampling sketching matrix, then for any matrix $\mathbf{B} = [\mathbf{b}_1, \ldots, \mathbf{b}_n] \in \mathbb{R}^{d \times n}$ with probability $1 - \delta$ for any $\delta > 0$ we have,*

$$\|\frac{1}{n}\mathbf{B}\mathbf{S}\mathbf{S}^\top\mathbf{1} - \frac{1}{n}\mathbf{B}\mathbf{1}\| \le (1 + \sqrt{2\ln(\frac{1}{\delta})})\sqrt{\frac{1}{s}}\max_i\|\mathbf{b}_i\|,$$

*where $\mathbf{1}$ is all ones vector.*

*Proof.* The vector $\mathbf{B}\mathbf{1}$ is the sum of column of the matrix $\mathbf{B}$ and $\mathbf{B}\mathbf{S}\mathbf{S}^\top\mathbf{1}$ is the sum of uniformly sampled and scaled column of the matrix $\mathbf{B}$ where the scaling factor is $\frac{1}{\sqrt{sp}}$ with $p = \frac{1}{n}$. If $(i_1, \ldots, i_s)$ is the set of sampled indices then $\mathbf{B}\mathbf{S}\mathbf{S}^\top\mathbf{1} = \sum_{k \in (i_1,\ldots,i_s)} \frac{1}{sp}\mathbf{b}_k$.

Define the function $f(i_1, \ldots, i_s) = \|\frac{1}{n}\mathbf{B}\mathbf{S}\mathbf{S}^\top\mathbf{1} - \frac{1}{n}\mathbf{B}\mathbf{1}\|$. Now consider a sampled set $(i_1, \ldots, i_{j'}, \ldots, i_s)$ with only one item (column) replaced then the bounded difference is

$$\Delta = |f(i_1, \ldots, i_j, \ldots, i_s) - f(i_1, \ldots, i_{j'}, \ldots, i_s)|$$
$$= |\frac{1}{n}\|\frac{1}{sp}\mathbf{b}_{i'_j} - \frac{1}{sp}\mathbf{b}_{i_j}\|\| \le \frac{2}{s}\max_i\|\mathbf{b}_i\|.$$

Now we have the expectation

$$\mathbb{E}[\|\frac{1}{n}\mathbf{B}\mathbf{S}\mathbf{S}^\top\mathbf{1} - \frac{1}{n}\mathbf{B}\mathbf{1}\|^2] \le \frac{n}{sn^2}\sum_{i=1}^n\|\mathbf{b}_i\|^2 = \frac{1}{s}\max_i\|\mathbf{b}_i\|^2$$

$$\Rightarrow \mathbb{E}[\|\frac{1}{n}\mathbf{B}\mathbf{S}\mathbf{S}^\top\mathbf{1} - \frac{1}{n}\mathbf{B}\mathbf{1}\|] \le \sqrt{\frac{1}{s}}\max_i\|\mathbf{b}_i\|.$$

459 Using McDiarmid inequality (Lemma 1) we have

$$P\left[\left\|\frac{1}{n}\mathbf{BSS}^\top\mathbf{1} - \frac{1}{n}\mathbf{B1}\right\| \geq \sqrt{\frac{1}{s}}\max_i \|\mathbf{b}_i\| + t\right] \leq \exp\left(-\frac{2t^2}{s\Delta^2}\right).$$

460 Equating the probability with $\delta$ we have

$$\exp(-\frac{2t^2}{s\Delta^2}) = \delta$$

$$\Rightarrow t = \Delta\sqrt{\frac{s}{2}\ln(\frac{1}{\delta})} = \max_i \|\mathbf{b}_i\|\sqrt{\frac{2}{s}\ln(\frac{1}{\delta})}.$$

461 Finally we have with probability $1 - \delta$

$$\|\frac{1}{n}\mathbf{BSS}^\top\mathbf{1} - \frac{1}{n}\mathbf{B1}\| \leq (1 + \sqrt{2\ln(\frac{1}{\delta})})\sqrt{\frac{1}{s}}\max_i \|\mathbf{b}_i\|.$$

462 $\qquad\square$

463 **Remark 8.** *For $m$ sketching matrix $\{\mathbf{S}_i\}_{i=1}^m$, the bound in the Lemma 3 is*

$$\|\frac{1}{n}\mathbf{BS}_i\mathbf{S}_i^\top\mathbf{1} - \frac{1}{n}\mathbf{B1}\| \leq (1 + \sqrt{2\ln(\frac{m}{\delta})})\sqrt{\frac{1}{s}}\max_i \|\mathbf{b}_i\|,$$

464 *with probability $1 - \delta$ for any $\delta > 0$ for all $i \in \{1, 2, \ldots, m\}$. In the case that each worker machine*
465 *holds data based on the uniform sketching matrix the local gradient is close to the exact gradient.*
466 *Thus the local second order update acts as a good approximate to the exact Newton update.*

467 Now we consider the update rule of GIANT [30] where the update is done in two rounds in each
468 iteration. In the first round each worker machine computes and send the local gradient and the
469 center machine computes the exact gradient $\mathbf{g}_t$ in iteration $t$. Next the center machine broadcasts
470 the exact gradient and each worker machine computes the local Hessian and send $\tilde{\mathbf{p}}_{i,t} = (\mathbf{H}_{i,t})^{-1}\mathbf{g}_t$
471 to the center machine and the center machine computes the approximate Newton direction $\tilde{\mathbf{p}}_t =$
472 $\frac{1}{m}\sum_{i=1}^m \tilde{\mathbf{p}}_{i,t}$. Now based on this we restate the following lemma (Lemma 6 [30]).

473 **Lemma 4.** *Let $\{\mathbf{S}_i\}_{i=1}^m \in \mathbb{R}^{n \times s}$ be sketching matrices based on Lemma 2. Let $\phi_t$ be defined in (10)*
474 *and $\tilde{\mathbf{p}}_t$ be the update. It holds that*

$$\min_\mathbf{p} \phi_t(\mathbf{p}) \leq \phi_t(\tilde{\mathbf{p}}_t) \leq (1 - \zeta^2)\min_\mathbf{p} \phi_t(\mathbf{p}),$$

475 *where $\zeta = \nu(\frac{\eta}{\sqrt{m}} + \frac{\eta^2}{1-\eta})$ and $\nu = \frac{\sigma_{max}(\mathbf{A}^\top\mathbf{A})}{\sigma_{max}(\mathbf{A}^\top\mathbf{A})+n\lambda} \leq 1$.*

476 Now we prove similar guarantee for the update according to COMRADE in Algorithm 1.

477 **Lemma 5.** *Let $\{\mathbf{S}_i\}_{i=1}^m \in \mathbb{R}^{n \times s}$ be sketching matrices based on Lemma 2. Let $\phi_t$ be defined in (10)*
478 *and $\hat{\mathbf{p}}_t$ be defined in Algorithm 1($\beta = 0$)*

$$\min_\mathbf{p} \phi_t(\mathbf{p}) \leq \phi_t(\hat{\mathbf{p}}_t) \leq \epsilon^2 + (1 - \zeta^2)\min_\mathbf{p} \phi_t(\mathbf{p}),$$

479 *where $\epsilon = \frac{1}{1-\eta}\frac{1}{\sqrt{\sigma_{min}(\mathbf{H}_t)}}(1 + \sqrt{2\ln(\frac{m}{\delta})})\sqrt{\frac{1}{s}}\max_i \|\mathbf{b}_i\|$ and $\zeta = \nu(\frac{\eta}{\sqrt{m}} + \frac{\eta^2}{1-\eta})$ and $\nu =$*
480 $\frac{\sigma_{max}(\mathbf{A}^\top\mathbf{A})}{\sigma_{max}(\mathbf{A}^\top\mathbf{A})+n\lambda}$.

481 *Proof.* First consider the quadratic function (10)

$$\phi_t(\hat{\mathbf{p}}_t) - \phi_t(\mathbf{p}^*) = \frac{1}{2}\|\mathbf{H}_t^{\frac{1}{2}}(\hat{\mathbf{p}}_t - \mathbf{p}^*)\|^2$$

$$\leq \underbrace{(\|\mathbf{H}_t^{\frac{1}{2}}(\hat{\mathbf{p}}_t - \tilde{\mathbf{p}}_t)\|^2}_{Term1} + \underbrace{\|\mathbf{H}_t^{\frac{1}{2}}(\tilde{\mathbf{p}}_t - \mathbf{p}^*)\|)^2}_{Term2}, \qquad (12)$$

482 where $\tilde{\mathbf{p}}_t = \frac{1}{m}\sum_{i=1}^m (\mathbf{H}_{i,t})^{-1}\mathbf{g}_t$. First we bound the Term 2 of (12) using the quadratic function and
483 Lemma 4

$$\frac{1}{2}\left\|\mathbf{H}_t^{\frac{1}{2}}(\tilde{\mathbf{p}}_t - \mathbf{p}^*)\|)^2 \leq \zeta^2 \left\|\mathbf{H}_t^{\frac{1}{2}}\mathbf{p}^*\right\|^2 \quad \text{(Using Lemma 4 )}$$

$$= -\zeta^2 \phi_t(\mathbf{p}^*). \tag{13}$$

484 The step in equation (13) is from the definition of the function $\phi_t$ and $\mathbf{p}^*$. It can be shown that

$$\phi_t(\mathbf{p}^*) = -\left\|\mathbf{H}_t^{\frac{1}{2}}\mathbf{p}^*\right\|^2.$$

485 Now we bound the Term 1 in (12). By Lemma 2, we have $(1-\eta)\mathbf{A}_t^\top\mathbf{A}_t \preceq \mathbf{A}_t^\top\mathbf{S}_i\mathbf{S}_i^\top\mathbf{A}_t \preceq$
486 $(1+\eta)\mathbf{A}_t^\top\mathbf{A}_t$. Following we have $(1-\eta)\mathbf{H}_t \preceq \mathbf{H}_{i,t} \preceq (1+\eta)\mathbf{H}_t$. Thus there exists matrix $\xi_i$
487 satisfying

$$\mathbf{H}_t^{\frac{1}{2}}\mathbf{H}_{i,t}^{-1}\mathbf{H}_t^{\frac{1}{2}} = \mathbf{I} + \xi_i \quad \text{and} \quad -\frac{\eta}{1+\eta} \preceq \xi_i \preceq \frac{\eta}{1-\eta},$$

488 So we have,

$$\left\|\mathbf{H}_t^{\frac{1}{2}}\mathbf{H}_{i,t}^{-1}\mathbf{H}_t^{\frac{1}{2}}\right\| \leq 1 + \frac{\eta}{1-\eta} = \frac{1}{1-\eta}. \tag{14}$$

489 Now we have

$$\left\|\mathbf{H}_t^{\frac{1}{2}}(\hat{\mathbf{p}}_t - \tilde{\mathbf{p}}_t)\right\| = \left\|\mathbf{H}_t^{\frac{1}{2}}\frac{1}{m}\sum_{i=1}^m (\hat{\mathbf{p}}_{i,t} - \tilde{\mathbf{p}}_{i,t})\right\|$$

$$\leq \frac{1}{m}\sum_{i=1}^m \left\|\mathbf{H}_t^{\frac{1}{2}}(\hat{\mathbf{p}}_{i,t} - \tilde{\mathbf{p}}_{i,t})\right\|$$

$$= \frac{1}{m}\sum_{i=1}^m \left\|\mathbf{H}_t^{\frac{1}{2}}\mathbf{H}_{i,t}^{-1}(\mathbf{g}_{i,t} - \mathbf{g}_t)\right\|$$

$$= \frac{1}{m}\sum_{i=1}^m \left\|\mathbf{H}_t^{\frac{1}{2}}\mathbf{H}_{i,t}^{-1}\mathbf{H}_t^{\frac{1}{2}}\mathbf{H}_t^{-\frac{1}{2}}(\mathbf{g}_{i,t} - \mathbf{g}_t)\right\|$$

$$\leq \frac{1}{m}\sum_{i=1}^m \left\|\mathbf{H}_t^{\frac{1}{2}}\mathbf{H}_{i,t}^{-1}\mathbf{H}_t^{\frac{1}{2}}\right\| \left\|\mathbf{H}_t^{-\frac{1}{2}}(\mathbf{g}_{i,t} - \mathbf{g}_t)\right\|$$

$$\leq \frac{1}{1-\eta}\frac{1}{m}\sum_{i=1}^m \left\|\mathbf{H}_t^{-\frac{1}{2}}(\mathbf{g}_{i,t} - \mathbf{g}_t)\right\| \quad \text{( Using (14))}$$

$$\leq \frac{1}{1-\eta}\frac{1}{\sqrt{\sigma_{min}(\mathbf{H}_t)}}\frac{1}{m}\sum_{i=1}^m \|(\mathbf{g}_{i,t} - \mathbf{g}_t)\|. \tag{15}$$

490 Now we bound $\|(\mathbf{g}_{i,t} - \mathbf{g}_t)\|$ using Lemma 3,

$$\|(\mathbf{g}_{i,t} - \mathbf{g}_t)\| = \|\frac{1}{n}\mathbf{B}\mathbf{S}\mathbf{S}^\top\mathbf{1} - \frac{1}{n}\mathbf{B}\mathbf{1}\| \leq (1 + \sqrt{2\ln(\frac{m}{\delta})})\sqrt{\frac{1}{s}}\max_i \|\mathbf{b}_i\|.$$

491 Plugging it into equation (15) we get,

$$\left\|\mathbf{H}_t^{\frac{1}{2}}(\hat{\mathbf{p}}_t - \tilde{\mathbf{p}}_t)\right\| \leq \frac{1}{1-\eta}\frac{1}{\sqrt{\sigma_{min}(\mathbf{H}_t)}}\frac{1}{m}\sum_{i=1}^m \|(\mathbf{g}_{i,t} - \mathbf{g}_t)\|$$

$$\leq \frac{1}{1-\eta}\frac{1}{\sqrt{\sigma_{min}(\mathbf{H}_t)}}(1 + \sqrt{2\ln(\frac{m}{\delta})})\sqrt{\frac{1}{s}}\max_i \|\mathbf{b}_i\|. \tag{16}$$

492 Now collecting the terms of (16) and (13) and plugging them into (12) we have

$$\phi_t(\hat{\mathbf{p}}_t) - \phi_t(\mathbf{p}^*) \leq \epsilon^2 - \zeta^2 \phi_t(\mathbf{p}^*)$$

$$\Rightarrow \phi_t(\hat{\mathbf{p}}_t) \leq \epsilon^2 + (1 - \zeta^2)\phi_t(\mathbf{p}^*),$$

493 where $\epsilon$ is as defined in (4).

494 $\qquad\qquad\qquad\qquad\qquad\qquad\qquad\qquad\qquad\qquad\qquad\qquad\qquad\qquad\qquad\qquad\qquad\qquad\qquad\square$

**Lemma 6.** *Let $\zeta \in (0,1), \epsilon$ be any fixed parameter. And $\hat{p}_t$ satisfies $\phi_t(\hat{\mathbf{p}}_t) \leq \epsilon^2 + (1 - \zeta^2) \min_{\mathbf{p}} \phi_t(\mathbf{p})$. Under the Assumption 1(Hessian L-Lipschitz) and $\boldsymbol{\Delta}_t = \mathbf{w}_t - \mathbf{w}^*$ satisfies*

$$\boldsymbol{\Delta}_{t+1}^\top \mathbf{H}_t \boldsymbol{\Delta}_{t+1} \leq L\|\boldsymbol{\Delta}_{t+1}\|\|\boldsymbol{\Delta}_t\|^2 + \frac{\zeta^2}{1-\zeta^2} \boldsymbol{\Delta}_t^\top \mathbf{H}_t \boldsymbol{\Delta}_t + 2\epsilon^2.$$

*Proof.* We have $\mathbf{w}_{t+1} = \mathbf{w}_t - \hat{\mathbf{p}}_t, \boldsymbol{\Delta}_t = \mathbf{w}_t - \mathbf{w}^*$ and $\boldsymbol{\Delta}_{t+1} = \mathbf{w}_{t+1} - \mathbf{w}^*$. Also $\hat{\mathbf{p}}_t = \mathbf{w}_t - \mathbf{w}_{t+1} = \boldsymbol{\Delta}_t - \boldsymbol{\Delta}_{t+1}$. From the definition of $\phi$ we have,

$$\phi_t(\hat{\mathbf{p}}_t) = \frac{1}{2}(\boldsymbol{\Delta}_t - \boldsymbol{\Delta}_{t+1})^\top \mathbf{H}_t (\boldsymbol{\Delta}_t - \boldsymbol{\Delta}_{t+1}) - (\boldsymbol{\Delta}_t - \boldsymbol{\Delta}_{t+1})) \, \mathbf{g}_t,$$

$$(1 - \zeta^2)\phi_t(\frac{1}{(1-\zeta^2)}\boldsymbol{\Delta}_t) = \frac{1}{2(1-\zeta^2)} \boldsymbol{\Delta}_t^\top \mathbf{H}_t \boldsymbol{\Delta}_t - \boldsymbol{\Delta}_t^\top \mathbf{g}_t.$$

From the above two equation we have

$$\phi_t(\hat{\mathbf{p}}_t) - (1 - \zeta^2)\phi_t(\frac{1}{(1-\zeta^2)}\boldsymbol{\Delta}_t)$$

$$= \frac{1}{2}\boldsymbol{\Delta}_{t+1}^\top \mathbf{H}_t \boldsymbol{\Delta}_{t+1} - \frac{1}{2}\boldsymbol{\Delta}_t^\top \mathbf{H}_t \boldsymbol{\Delta}_{t+1} + \frac{1}{2}\boldsymbol{\Delta}_{t+1}^\top \mathbf{g}_t - \frac{\zeta^2}{2(1-\zeta^2)} \boldsymbol{\Delta}_t^\top \mathbf{H}_t \boldsymbol{\Delta}_t.$$

From Lemma 5 the following holds

$$\phi_t(\hat{\mathbf{p}}_t) \leq \epsilon^2 + (1 - \zeta^2) \min_{\mathbf{p}} \phi_t(\mathbf{p})$$

$$\leq \epsilon^2 + (1 - \zeta^2)\phi_t(\frac{1}{(1-\zeta^2)}\boldsymbol{\Delta}_t).$$

So we have

$$\frac{1}{2}\boldsymbol{\Delta}_{t+1}^\top \mathbf{H}_t \boldsymbol{\Delta}_{t+1} - \boldsymbol{\Delta}_t^\top \mathbf{H}_t \boldsymbol{\Delta}_{t+1} + \boldsymbol{\Delta}_{t+1}^\top \mathbf{g}_t - \frac{\zeta^2}{2(1-\zeta^2)} \boldsymbol{\Delta}_t^\top \mathbf{H}_t \boldsymbol{\Delta}_t \leq \epsilon^2. \qquad (17)$$

Consider $\mathbf{g}_t = \mathbf{g}(\mathbf{w}_t)$

$$\mathbf{g}(\mathbf{w}_t) = \mathbf{g}(\mathbf{w}^*) + \left(\int_0^1 \nabla^2 f(\mathbf{w}^* + z(\mathbf{w}_t - \mathbf{w}^*))dz\right)(\mathbf{w}_t - \mathbf{w}^*)$$

$$= \left(\int_0^1 \nabla^2 f(\mathbf{w}^* + z(\mathbf{w}_t - \mathbf{w}^*))dz\right)\boldsymbol{\Delta}_t \quad (\text{as } \mathbf{g}(\mathbf{w}^*) = 0).$$

Now we bound the following

$$\|\mathbf{H}_t\boldsymbol{\Delta}_t - \mathbf{g}(\mathbf{w}_t)\| \leq \|\boldsymbol{\Delta}_t\| \left\|\int_0^1 [\nabla^2 f(\mathbf{w}_t) - \nabla^2 f(\mathbf{w}^* + z(\mathbf{w}_t - \mathbf{w}^*))]dz\right\|$$

$$\leq \|\boldsymbol{\Delta}_t\| \int_0^1 \left\|[\nabla^2 f(\mathbf{w}_t) - \nabla^2 f(\mathbf{w}^* + z(\mathbf{w}_t - \mathbf{w}^*))]\right\| dz \quad \text{(By Jensen's Inequality)}$$

$$\leq \|\boldsymbol{\Delta}_t\| \int_0^1 (1-z)L \|\mathbf{w}_t - \mathbf{w}^*\| dz \quad \text{(by } L\text{-Lipschitz assumption)}$$

$$= \frac{L}{2} \|\boldsymbol{\Delta}_t\|^2.$$

Plugging it into (17) we have

$$\boldsymbol{\Delta}_{t+1}^\top \mathbf{H}_t \boldsymbol{\Delta}_{t+1} \leq 2\boldsymbol{\Delta}_{t+1}^\top (\mathbf{H}_t\boldsymbol{\Delta}_t - \mathbf{g}_t) + \frac{\zeta^2}{(1-\zeta^2)} \boldsymbol{\Delta}_t^\top \mathbf{H}_t \boldsymbol{\Delta}_t + 2\epsilon^2$$

$$\leq 2\|\boldsymbol{\Delta}_{t+1}\| \|\mathbf{H}_t\boldsymbol{\Delta}_t - \mathbf{g}_t\| + \frac{\zeta^2}{(1-\zeta^2)} \boldsymbol{\Delta}_t^\top \mathbf{H}_t \boldsymbol{\Delta}_t + 2\epsilon^2$$

$$\leq L \|\boldsymbol{\Delta}_{t+1}\| \|\boldsymbol{\Delta}_t\|^2 + \frac{\zeta^2}{(1-\zeta^2)} \boldsymbol{\Delta}_t^\top \mathbf{H}_t \boldsymbol{\Delta}_t + 2\epsilon^2.$$

$\square$

**Proof of Theorem 1**

*Proof.* From the Lemma 6 with probability $1 - \delta$

$$\boldsymbol{\Delta}_{t+1}^{\top}\mathbf{H}_t\boldsymbol{\Delta}_{t+1} \leq L\,\|\boldsymbol{\Delta}_{t+1}\|\,\|\boldsymbol{\Delta}_t\|^2 + \frac{\zeta^2}{(1-\zeta^2)}\boldsymbol{\Delta}_t^{\top}\mathbf{H}_t\boldsymbol{\Delta}_t + 2\epsilon^2$$

$$\leq L\|\boldsymbol{\Delta}_{t+1}\|\|\boldsymbol{\Delta}_t\|^2 + (\frac{\zeta^2}{1-\zeta^2}\sigma_{max}(\mathbf{H}_t))\|\boldsymbol{\Delta}_t\|^2 + 2\epsilon^2.$$

So we have,

$$\|\boldsymbol{\Delta}_{t+1}\| \leq \max\{\sqrt{\frac{\sigma_{max}(\mathbf{H}_t)}{\sigma_{min}(\mathbf{H}_t)}(\frac{\zeta^2}{1-\zeta^2})}\|\boldsymbol{\Delta}_t\|, \frac{L}{\sigma_{min}(\mathbf{H}_t)}\|\boldsymbol{\Delta}_t\|^2\} + \frac{2\epsilon}{\sqrt{\sigma_{min}(\mathbf{H}_t)}}.$$

$\square$

# 9    Appendix B: Analysis of Section 4

In this section we provide the theoretical analysis of the Byzantine robust method explained in Section 4 and prove the statistical guarantee. In any iteration $t$ the following holds

$$|\mathcal{U}_t| = |(\mathcal{U}_t \cap \mathcal{M}_t)| + |(\mathcal{U}_t \cap \mathcal{B}_t)|$$
$$|\mathcal{M}_t| = |(\mathcal{U}_t \cap \mathcal{M}_t)| + |(\mathcal{M}_t \cap \mathcal{T}_t)|.$$

Combining both we have

$$|\mathcal{U}_t| = |\mathcal{M}_t| - |(\mathcal{M}_t \cap \mathcal{T}_t)| + |(\mathcal{U}_t \cap \mathcal{B}_t)|.$$

**Lemma 7.** *Let $\{\mathbf{S}_i\}_{i=1}^m \in \mathbb{R}^{n\times s}$ be sketching matrices based on Lemma 2. Let $\phi_t$ be defined in* (10) *and $\hat{\mathbf{p}}_t$ be defined in Algorithm 1. It holds that*

$$\min_{\mathbf{p}} \phi_t(\mathbf{p}) \leq \phi_t(\hat{\mathbf{p}}_t) \leq \epsilon_{byz}^2 + (1 - \zeta_{byz}^2)\phi(\mathbf{p}^*),$$

*where $\epsilon_{byz}$ and $\zeta_{byz}$ is defined in* (5) *and* (6) *respectively.*

*Proof.* In the following analysis we omit the subscript '$t$'. From the definition of the quadratic function (10) we know that

$$\phi(\hat{\mathbf{p}}) - \phi(\mathbf{p}^*) = \frac{1}{2}\|\mathbf{H}^{\frac{1}{2}}(\hat{\mathbf{p}} - \mathbf{p}^*)\|^2.$$

Now we consider

$$\frac{1}{2}\|\mathbf{H}^{\frac{1}{2}}(\hat{\mathbf{p}} - \mathbf{p}^*)\|^2 = \frac{1}{2}\|\mathbf{H}^{\frac{1}{2}}(\frac{1}{|\mathcal{U}|}\sum_{i\in\mathcal{U}}\hat{\mathbf{p}}_i - \mathbf{p}^*)\|^2$$

$$= \frac{1}{2}\|\mathbf{H}^{\frac{1}{2}}\frac{1}{|\mathcal{U}|}(\sum_{i\in\mathcal{M}}(\hat{\mathbf{p}}_i - \mathbf{p}^*) - \sum_{i\in(\mathcal{M}\cap\mathcal{T})}(\hat{\mathbf{p}}_i - \mathbf{p}^*) + \sum_{i\in(\mathcal{U}\cap\mathcal{B})}(\hat{\mathbf{p}}_i - \mathbf{p}^*))\|^2$$

$$\leq \underbrace{\|\mathbf{H}^{\frac{1}{2}}\frac{1}{|\mathcal{U}|}(\sum_{i\in\mathcal{M}}(\hat{\mathbf{p}}_i - \mathbf{p}^*)\|^2}_{Term1} + \underbrace{2\|\mathbf{H}^{\frac{1}{2}}\frac{1}{|\mathcal{U}|}\sum_{i\in(\mathcal{M}\cap\mathcal{T})}(\hat{\mathbf{p}}_i - \mathbf{p}^*)\|^2}_{Term2} + \underbrace{2\|\mathbf{H}^{\frac{1}{2}}\frac{1}{|\mathcal{U}|}\sum_{i\in(\mathcal{U}\cap\mathcal{B})}(\hat{\mathbf{p}}_i - \mathbf{p}^*))\|^2}_{Term3}.$$

Now we bound each term separately and use the result of the Lemma 5 to bound each term.

$$Term1 = \|\mathbf{H}^{\frac{1}{2}}\frac{1}{|\mathcal{U}|}(\sum_{i\in\mathcal{M}}(\hat{\mathbf{p}}_i - \mathbf{p}^*)\|^2$$

$$= (\frac{1-\alpha}{1-\beta})^2\|\mathbf{H}^{\frac{1}{2}}\frac{1}{|\mathcal{M}|}(\sum_{i\in\mathcal{M}}(\hat{\mathbf{p}}_i - \mathbf{p}^*)\|^2$$

$$\leq (\frac{1-\alpha}{1-\beta})^2[\epsilon^2 + \zeta_{\mathcal{M}}^2\|\mathbf{H}^{\frac{1}{2}}\mathbf{p}^*\|^2],$$

521    where $\zeta_{\mathcal{M}} = \nu\left(\frac{\eta}{\sqrt{|\mathcal{M}|}} + \frac{\eta^2}{1-\eta}\right) = \nu\left(\frac{\eta}{\sqrt{(1-\alpha)m}} + \frac{\eta^2}{1-\eta}\right).$

522    Similarly the Term 2 can be bonded as it is a bound on good machines

$$Term2 = 2\|\mathbf{H}^{\frac{1}{2}}\frac{1}{|\mathcal{U}|}\sum_{i \in (\mathcal{M} \cap \mathcal{T})}(\hat{\mathbf{p}}_i - \mathbf{p}^*)\|^2$$

$$= 2\left(\frac{1-\alpha}{1-\beta}\right)^2\|\mathbf{H}^{\frac{1}{2}}\frac{1}{|\mathcal{M} \cap \mathcal{T}|}\sum_{i \in (\mathcal{M} \cap \mathcal{T})}(\hat{\mathbf{p}}_i - \mathbf{p}^*)\|^2$$

$$\leq 2\left(\frac{1-\alpha}{1-\beta}\right)^2[\epsilon^2 + \zeta_{\mathcal{M} \cap \mathcal{T}}^2\|\mathbf{H}^{\frac{1}{2}}\mathbf{p}^*\|^2],$$

523    where $\zeta_{\mathcal{M} \cap \mathcal{T}} = \nu\left(\frac{\eta}{\sqrt{|\mathcal{M} \cap \mathcal{T}|}} + \frac{\eta^2}{1-\eta}\right) \leq \nu\left(\frac{\eta}{\sqrt{(1-\beta)m}} + \frac{\eta^2}{1-\eta}\right).$

524    For the Term 3 we know that $\beta > \alpha$ so all the untrimmed worker norm is bounded by a good machine
525    as at least one good machine gets trimmed.

$$Term3 = 2\|\mathbf{H}^{\frac{1}{2}}\frac{1}{|\mathcal{U}|}\sum_{i \in (\mathcal{U} \cap \mathcal{B})}(\hat{\mathbf{p}}_i - \mathbf{p}^*))\|^2$$

$$\leq 2\sigma_{max}(\mathbf{H})\left(\frac{|\mathcal{U} \cap \mathcal{B}|}{|\mathcal{U}|}\right)^2\|\frac{1}{|\mathcal{U} \cap \mathcal{B}|}\sum_{i \in (\mathcal{U} \cap \mathcal{B})}(\hat{\mathbf{p}}_i - \mathbf{p}^*))\|^2$$

$$\leq 2\sigma_{max}(\mathbf{H})\left(\frac{|\mathcal{U} \cap \mathcal{B}|}{|\mathcal{U}|}\right)^2\frac{1}{|\mathcal{U} \cap \mathcal{B}|}\sum_{i \in (\mathcal{U} \cap \mathcal{B})}\|(\hat{\mathbf{p}}_i - \mathbf{p}^*))\|^2$$

$$\leq 4\sigma_{max}(\mathbf{H})\left(\frac{|\mathcal{U} \cap \mathcal{B}|}{|\mathcal{U}|}\right)^2\frac{1}{|\mathcal{U} \cap \mathcal{B}|}\sum_{i \in (\mathcal{U} \cap \mathcal{B})}(\|\hat{\mathbf{p}}_i\|^2 + \|\mathbf{p}^*\|^2)$$

$$\leq 4\sigma_{max}(\mathbf{H})\left(\frac{|\mathcal{U} \cap \mathcal{B}|}{|\mathcal{U}|}\right)^2\max_{i \in \mathcal{M}}(\|\hat{\mathbf{p}}_i\|^2 + \|\mathbf{p}^*\|^2)$$

$$\leq 4\sigma_{max}(\mathbf{H})\left(\frac{|\mathcal{U} \cap \mathcal{B}|}{|\mathcal{U}|}\right)^2\max_{i \in \mathcal{M}}(\|\hat{\mathbf{p}}_i - \mathbf{p}^*\|^2 + 2\|\mathbf{p}^*\|^2)$$

$$\leq 4\kappa\left(\frac{|\mathcal{U} \cap \mathcal{B}|}{|\mathcal{U}|}\right)^2\max_{i \in \mathcal{M}}(\|\mathbf{H}^{\frac{1}{2}}(\hat{\mathbf{p}}_i - \mathbf{p}^*)\|^2 + 2\|\mathbf{H}^{\frac{1}{2}}\mathbf{p}^*\|^2)$$

$$\leq 4\kappa\left(\frac{|\mathcal{U} \cap \mathcal{B}|}{|\mathcal{U}|}\right)^2(\epsilon^2 + (2 + \zeta_1^2)\|\mathbf{H}^{\frac{1}{2}}\mathbf{p}^*\|^2)$$

$$\leq 4\kappa\left(\frac{\alpha}{1-\beta}\right)^2(\epsilon^2 + (2 + \zeta_1^2)\|\mathbf{H}^{\frac{1}{2}}\mathbf{p}^*\|^2),$$

526    where $\zeta_1 = \nu(\eta + \frac{\eta^2}{1-\eta}) = \frac{\nu}{1-\eta}$ and $\kappa = \frac{\sigma_{max}(\mathbf{H})}{\sigma_{min}(\mathbf{H})}.$

527    Combining all the bounds on Term1 , Term2 and Term3 we have

$$\frac{1}{2}\|\mathbf{H}^{\frac{1}{2}}(\hat{\mathbf{p}} - \mathbf{p}^*)\|^2 \leq \epsilon_{byz}^2 + \zeta_{byz}^2\|\mathbf{H}^{\frac{1}{2}}\mathbf{p}^*\|^2,$$

528    where

$$\epsilon_{byz}^2 = \left(3\left(\frac{1-\alpha}{1-\beta}\right)^2 + 4\kappa\left(\frac{\alpha}{1-\beta}\right)^2\right)\epsilon^2,$$

$$\zeta_{byz}^2 = 2\left(\frac{1-\alpha}{1-\beta}\right)^2\zeta_{\mathcal{M} \cap \mathcal{T}}^2 + \left(\frac{1-\alpha}{1-\beta}\right)^2\zeta_{\mathcal{M}}^2 + 4\kappa\left(\frac{\alpha}{1-\beta}\right)^2(2 + \zeta_1^2).$$

529    Finally we have

$$\phi(\hat{\mathbf{p}}) - \phi(\mathbf{p}^*) \leq \epsilon_{byz}^2 - \zeta_{byz}^2\phi(\mathbf{p}^*)$$

$$\Rightarrow \phi(\hat{\mathbf{p}}) \leq \epsilon_{byz}^2 + (1 - \zeta_{byz}^2)\phi(\mathbf{p}^*).$$

530                                                             $\square$

**Lemma 8.** *Let $\zeta_{byz} \in (0,1)$, $\epsilon_{byz}$ be any fixed parameter. And $\hat{p}_t$ satisfies $\phi_t(\hat{\mathbf{p}}_t) \leq \epsilon_{byz}^2 + (1 - \zeta_{byz}^2) \min_{\mathbf{p}} \phi_t(\mathbf{p})$. Under the Assumption 1(Hessian L-Lipschitz) and $\mathbf{\Delta}_t = \mathbf{w}_t - \mathbf{w}^*$ satisfies*

$$\mathbf{\Delta}_{t+1}^T \mathbf{H}_t \mathbf{\Delta}_{t+1} \leq L \|\mathbf{\Delta}_{t+1}\| \|\mathbf{\Delta}_t\|^2 + \frac{\zeta_{byz}^2}{1 - \zeta_{byz}^2} \mathbf{\Delta}_t^T \mathbf{H}_t \mathbf{\Delta}_t + 2\epsilon_{byz}^2.$$

*Proof.* We choose $\zeta = \zeta_{byz}$ and $\epsilon = \epsilon_{byz}$ from the Lemma 7 and follow the proof of Lemma 6 to obtain the desired bound. □

**Proof of Theorem 2**

*Proof.* We get the desired bound by developing from the result of the Lemma 8 and following the proof of Theorem 1 □

# 10    Appendix C:Analysis of Section 5

First we prove the following lemma that will be useful in our subsequent calculations. Consider that $\mathcal{Q}(\hat{\mathbf{p}}) = \frac{1}{|B|} \sum_{i \in B} \mathcal{Q}(\hat{\mathbf{p}}_i)$. And also we use the following notation $\zeta_B = \nu(\frac{\eta}{\sqrt{|B|}} + \frac{\eta^2}{1 - \eta})$, $\nu = \frac{\sigma_{max}(\mathbf{A}^\top \mathbf{A})}{\sigma_{max}(\mathbf{A}^\top \mathbf{A}) + n\lambda} \leq 1$.

**Lemma 9.** *If $\mathcal{Q}(\hat{\mathbf{p}}_i)$ is the local update direction and $\mathbf{p}^*$ is the optimal solution to the quadratic function $\phi$ then*

$$\left\| \mathbf{H}^{\frac{1}{2}} (\mathcal{Q}(\hat{\mathbf{p}}_i) - \mathbf{p}^*) \right\|^2 \leq 1 + \kappa(1 - \rho))\epsilon^2 + (\zeta_B^2 + \kappa(1 - \rho)((1 + \zeta_1^2)) \left\| \mathbf{H}^{\frac{1}{2}} \mathbf{p}^* \right\|^2,$$

*where $\mathbf{H}$ is the exact Hessian and*

$$\epsilon_1 = \sqrt{(1 + \kappa(1 - \rho))}\epsilon,$$
$$\zeta_{comp,B}^2 = (\zeta_B^2 + \kappa(1 - \rho)((1 + \zeta_1^2)).$$

*$\epsilon$ is defined in equation (4) and*

*Proof.*

$$\left\| \mathbf{H}^{\frac{1}{2}} (\mathcal{Q}(\hat{\mathbf{p}}) - \mathbf{p}^*) \right\|^2 = \left\| \mathbf{H}^{\frac{1}{2}} (\mathcal{Q}(\hat{\mathbf{p}}) - \hat{\mathbf{p}} + \hat{\mathbf{p}} - \mathbf{p}^*) \right\|^2$$

$$\leq 2 \left( \underbrace{\left\| \mathbf{H}^{\frac{1}{2}} (\mathcal{Q}(\hat{\mathbf{p}}) - \hat{\mathbf{p}}) \right\|^2}_{Term1} + \underbrace{\left\| \mathbf{H}^{\frac{1}{2}} (\hat{\mathbf{p}} - \mathbf{p}^*) \right\|^2}_{Term2} \right). \quad (18)$$

Following the proof of Lemma 5 we get

$$\left\| \mathbf{H}^{\frac{1}{2}} (\hat{\mathbf{p}}_i - \mathbf{p}^*) \right\|^2 \leq \epsilon^2 + \zeta_1 \left\| \mathbf{H}^{\frac{1}{2}} \mathbf{p}^* \right\|^2, \quad (19)$$

where $\epsilon$ is as defined in (4).Now we consider the term

$$\left\| \mathbf{H}^{\frac{1}{2}} (\mathcal{Q}(\hat{\mathbf{p}}_i) - \hat{\mathbf{p}}_i) \right\|^2 \leq \sigma_{max}(\mathbf{H})(1 - \rho) \|\hat{\mathbf{p}}_i\|^2$$

$$\leq \sigma_{max}(\mathbf{H})(1 - \rho) \left( \|\hat{\mathbf{p}}_i - \mathbf{p}^*\|^2 + \|\mathbf{p}^*\|^2 \right)$$

$$\leq \frac{\sigma_{max}}{\sigma_{min}} (1 - \rho) \left( \left\| \mathbf{H}^{\frac{1}{2}} (\hat{\mathbf{p}}_i - \mathbf{p}^*) \right\|^2 + \left\| \mathbf{H}^{\frac{1}{2}} \mathbf{p}^* \right\|^2 \right)$$

$$= \kappa(1 - \rho) \left( \left\| \mathbf{H}^{\frac{1}{2}} (\hat{\mathbf{p}}_i - \mathbf{p}^*) \right\|^2 + \left\| \mathbf{H}^{\frac{1}{2}} \mathbf{p}^* \right\|^2 \right)$$

$$\leq \kappa(1 - \rho) \left( \epsilon^2 + (1 + \zeta_1^2) \left\| \mathbf{H}^{\frac{1}{2}} \mathbf{p}^* \right\|^2 \right) \quad \text{Using (19).}$$

548 Now we use the above calculation and bound Term1

$$\left\|\mathbf{H}^{\frac{1}{2}}(\mathcal{Q}(\hat{\mathbf{p}}) - \hat{\mathbf{p}})\right\|^2 \leq \frac{1}{|B|} \sum_{i \in B} \left\|\mathbf{H}^{\frac{1}{2}}(\mathcal{Q}(\hat{\mathbf{p}}_i) - \hat{\mathbf{p}}_i)\right\|^2$$

$$\leq \kappa(1 - \rho)\left(\epsilon^2 + (1 + \zeta_1^2)\left\|\mathbf{H}^{\frac{1}{2}}\mathbf{p}^*\right\|^2\right). \tag{20}$$

549 We can bound the Term2 directly using the proof of Lemma 5

$$\left\|\mathbf{H}^{\frac{1}{2}}(\hat{\mathbf{p}} - \mathbf{p}^*)\right\|^2 \leq \epsilon^2 + \zeta_B^2 \left\|\mathbf{H}^{\frac{1}{2}}\mathbf{p}^*\right\|^2. \tag{21}$$

550 Now we use (20) and (21) and plug them in (18)

$$\left\|\mathbf{H}^{\frac{1}{2}}(\mathcal{Q}(\hat{\mathbf{p}}) - \mathbf{p}^*)\right\|^2 \leq (1 + \kappa(1 - \rho))\epsilon^2 + (\zeta_B^2 + \kappa(1 - \rho)((1 + \zeta_1^2)))\left\|\mathbf{H}^{\frac{1}{2}}\mathbf{p}^*\right\|^2.$$

551 Now we define

$$\epsilon_1 = \sqrt{(1 + \kappa(1 - \rho))}\epsilon$$
$$\zeta_{comp,B}^2 = (\zeta_B^2 + \kappa(1 - \rho)((1 + \zeta_1^2)).$$

552 □

553 Now we have the robust update in iteration $t$ to be $\mathcal{Q}(\hat{\mathbf{p}}) = \frac{1}{|\mathcal{U}_t|} \sum_{i \in \mathcal{U}_t} \mathcal{Q}(\hat{p}_{i,t})$.

554 **Lemma 10.** *Let* $\{\mathbf{S}_i\}_{i=1}^m \in \mathbb{R}^{n \times s}$ *be sketching matrices based on Lemma 2. Let* $\phi_t$ *be defined in*
555 (10) *and* $\mathcal{Q}(\hat{\mathbf{p}}_t)$ *be the update with* $\mathcal{Q}$ *being* $\rho$*-approximate compressor. It holds that*

$$\min_{\mathbf{p}} \phi_t(\mathbf{p}) \leq \phi_t(\mathcal{Q}(\hat{\mathbf{p}}_t)) \leq \epsilon_{comp,byz}^2 + (1 - \zeta_{comp,byz}^2)\phi_t(\mathbf{p}^*),$$

556 *where* $\epsilon_{comp,byz}$ *and* $\zeta_{comp,byz}^2$ *is as defined in* (7) *and* (8) *respectively.*

557 *Proof.* In the following analysis we omit the subscript '$t$'. From the definition of the quadratic
558 function (10) we know that

$$\phi(\mathcal{Q}(\hat{\mathbf{p}})) - \phi(\mathbf{p}^*) = \frac{1}{2}\|\mathbf{H}^{\frac{1}{2}}(\mathcal{Q}(\hat{\mathbf{p}}) - \mathbf{p}^*)\|^2.$$

559 Now we consider
$$\frac{1}{2}\|\mathbf{H}^{\frac{1}{2}}(\mathcal{Q}(\hat{\mathbf{p}}) - \mathbf{p}^*)\|^2 = \frac{1}{2}\|\mathbf{H}^{\frac{1}{2}}(\frac{1}{|\mathcal{U}|}\sum_{i \in \mathcal{U}} \mathcal{Q}(\hat{\mathbf{p}}_i) - \mathbf{p}^*)\|^2$$

$$= \frac{1}{2}\|\mathbf{H}^{\frac{1}{2}}\frac{1}{|\mathcal{U}|}(\sum_{i \in \mathcal{M}}(\mathcal{Q}(\hat{\mathbf{p}}_i) - \mathbf{p}^*) - \sum_{i \in (\mathcal{M} \cap \mathcal{T})}(\mathcal{Q}(\hat{\mathbf{p}}_i) - \mathbf{p}^*) + \sum_{i \in (\mathcal{U} \cap \mathcal{B})}(\mathcal{Q}(\hat{\mathbf{p}}_i) - \mathbf{p}^*))\|^2$$

$$\leq \underbrace{\|\mathbf{H}^{\frac{1}{2}}\frac{1}{|\mathcal{U}|}(\sum_{i \in \mathcal{M}}(\mathcal{Q}(\hat{\mathbf{p}}_i) - \mathbf{p}^*)\|^2}_{Term1} + \underbrace{2\|\mathbf{H}^{\frac{1}{2}}\frac{1}{|\mathcal{U}|}\sum_{i \in (\mathcal{M} \cap \mathcal{T})}(\mathcal{Q}(\hat{\mathbf{p}}_i) - \mathbf{p}^*)\|^2}_{Term2}$$

$$+ \underbrace{2\|\mathbf{H}^{\frac{1}{2}}\frac{1}{|\mathcal{U}|}\sum_{i \in (\mathcal{U} \cap \mathcal{B})}(\mathcal{Q}(\hat{\mathbf{p}}_i) - \mathbf{p}^*))\|^2}_{Term3}.$$

560 Now we bound each term separately and use the Lemma 9

$$Term1 = \|\mathbf{H}^{\frac{1}{2}}\frac{1}{|\mathcal{U}|}(\sum_{i \in \mathcal{M}}(\mathcal{Q}(\hat{\mathbf{p}}_i) - \mathbf{p}^*)\|^2$$

$$= (\frac{1 - \alpha}{1 - \beta})^2\|\mathbf{H}^{\frac{1}{2}}\frac{1}{|\mathcal{M}|}(\sum_{i \in \mathcal{M}}(\mathcal{Q}(\hat{\mathbf{p}}_i) - \mathbf{p}^*)\|^2$$

$$\leq (\frac{1 - \alpha}{1 - \beta})^2[\epsilon_1^2 + \zeta_{comp,\mathcal{M}}^2\|\mathbf{H}^{\frac{1}{2}}\mathbf{p}^*\|^2],$$

561 where $\zeta_{comp,\mathcal{M}}^2 = (\zeta_{\mathcal{M}}^2 + \kappa(1-\rho))((1+\zeta_1^2)$. Similarly the Term 2 can be bonded as it is a bound on
562 good machines

$$Term2 = 2\|\mathbf{H}^{\frac{1}{2}} \frac{1}{|\mathcal{U}|} \sum_{i\in(\mathcal{M}\cap\mathcal{T})} (\mathcal{Q}(\hat{\mathbf{p}}_i) - \mathbf{p}^*)\|^2$$

$$= 2(\frac{1-\alpha}{1-\beta})^2 \|\mathbf{H}^{\frac{1}{2}} \frac{1}{|\mathcal{M}\cap\mathcal{T}|} \sum_{i\in(\mathcal{M}\cap\mathcal{T})} (\mathcal{Q}(\hat{\mathbf{p}}_i) - \mathbf{p}^*)\|^2$$

$$\leq 2(\frac{1-\alpha}{1-\beta})^2 [\epsilon_1^2 + \zeta_{comp,\mathcal{M}\cap\mathcal{T}}^2 \|\mathbf{H}^{\frac{1}{2}} \mathbf{p}^*\|^2].$$

563 For the Term 3 we know that $\beta > \alpha$ so all the untrimmed worker norm is bounded by a good machine
564 as at least one good machine gets trimmed.

$$Term3 = 2\|\mathbf{H}^{\frac{1}{2}} \frac{1}{|\mathcal{U}|} \sum_{i\in(\mathcal{U}\cap\mathcal{B})} (\mathcal{Q}(\hat{\mathbf{p}}_i) - \mathbf{p}^*))\|^2$$

$$\leq 2\sigma_{max}(\mathbf{H})(\frac{|\mathcal{U}\cap\mathcal{B}|}{|\mathcal{U}|})^2 \|\frac{1}{|\mathcal{U}\cap\mathcal{B}|} \sum_{i\in(\mathcal{U}\cap\mathcal{B})} (\mathcal{Q}(\hat{\mathbf{p}}_i) - \mathbf{p}^*))\|^2$$

$$\leq 2\sigma_{max}(\mathbf{H})(\frac{|\mathcal{U}\cap\mathcal{B}|}{|\mathcal{U}|})^2 \frac{1}{|\mathcal{U}\cap\mathcal{B}|} \sum_{i\in(\mathcal{U}\cap\mathcal{B})} \|(\mathcal{Q}(\hat{\mathbf{p}}_i) - \mathbf{p}^*))\|^2$$

$$\leq 4\sigma_{max}(\mathbf{H})(\frac{|\mathcal{U}\cap\mathcal{B}|}{|\mathcal{U}|})^2 \frac{1}{|\mathcal{U}\cap\mathcal{B}|} \sum_{i\in(\mathcal{U}\cap\mathcal{B})} (\|\mathcal{Q}(\hat{\mathbf{p}}_i)\|^2 + \|\mathbf{p}^*\|^2)$$

$$\leq 4\sigma_{max}(\mathbf{H})(\frac{|\mathcal{U}\cap\mathcal{B}|}{|\mathcal{U}|})^2 \max_{i\in\mathcal{M}}(\|\mathcal{Q}(\hat{\mathbf{p}}_i)\|^2 + \|\mathbf{p}^*\|^2)$$

$$\leq 4\sigma_{max}(\mathbf{H})(\frac{|\mathcal{U}\cap\mathcal{B}|}{|\mathcal{U}|})^2 \max_{i\in\mathcal{M}}(\|\mathcal{Q}(\hat{\mathbf{p}}_i) - \mathbf{p}^*\|^2 + 2\|\mathbf{p}^*\|^2)$$

$$\leq 4\kappa(\frac{|\mathcal{U}\cap\mathcal{B}|}{|\mathcal{U}|})^2 \max_{i\in\mathcal{M}}(\|\mathbf{H}^{\frac{1}{2}}(\mathcal{Q}(\hat{\mathbf{p}}_i) - \mathbf{p}^*)\|^2 + 2\|\mathbf{H}^{\frac{1}{2}}\mathbf{p}^*\|^2)$$

$$\leq 4\kappa(\frac{|\mathcal{U}\cap\mathcal{B}|}{|\mathcal{U}|})^2 (\epsilon_1^2 + (2 + \zeta_1^2)\|\mathbf{H}^{\frac{1}{2}}\mathbf{p}^*\|^2)$$

$$\leq 4\kappa(\frac{\alpha}{1-\beta})^2 (\epsilon_1^2 + (2 + \zeta_1^2)\|\mathbf{H}^{\frac{1}{2}}\mathbf{p}^*\|^2).$$

565 Combining all the bounds on Term1 , Term2 and Term3 we have

$$\frac{1}{2}\|\mathbf{H}^{\frac{1}{2}}(\hat{\mathbf{p}} - \mathbf{p}^*)\|^2 \leq \epsilon_{byz}^2 + \zeta_{byz}^2\|\mathbf{H}^{\frac{1}{2}}\mathbf{p}^*\|^2,$$

566 where

$$\epsilon_{comp,byz}^2 = \left( 3\left(\frac{1-\alpha}{1-\beta}\right)^2 + 4\kappa\left(\frac{\alpha}{1-\beta}\right)^2 \right)\epsilon_1^2$$

$$\zeta_{comp,byz}^2 = 2\left(\frac{1-\alpha}{1-\beta}\right)^2 \zeta_{comp,\mathcal{M}\cap\mathcal{T}}^2 + \left(\frac{1-\alpha}{1-\beta}\right)^2 \zeta_{comp,\mathcal{M}}^2 + 4\kappa\left(\frac{\alpha}{1-\beta}\right)^2 (2 + \zeta_{comp,1}^2).$$

567 Finally we have

$$\phi(\hat{\mathbf{p}}) - \phi(\mathbf{p}^*) \leq \epsilon_{comp,byz}^2 - \zeta_{comp,byz}^2\phi(\mathbf{p}^*)$$
$$\Rightarrow \phi(\hat{\mathbf{p}}) \leq \epsilon_{comp,byz}^2 + (1 - \zeta_{comp,byz}^2)\phi(\mathbf{p}^*).$$

568 $\qquad\qquad\qquad\qquad\qquad\qquad\qquad\qquad\qquad\qquad\qquad\qquad\qquad\qquad\qquad\qquad\square$

569 **Lemma 11.** *Let* $\zeta_{comp,byz} \in (0,1), \epsilon_{comp,byz}$ *be any fixed parameter. And* $\mathcal{Q}(\hat{p}_t)$ *satisfies*
570 $\phi_t(\mathcal{Q}(\hat{p}_t)) \leq \epsilon_{byz}^2 + (1 - \zeta_{byz}^2) \min_{\mathbf{p}} \phi_t(\mathbf{p})$. *Under the Assumption 1(Hessian L-Lipschitz) and*
571 $\boldsymbol{\Delta}_t = \mathbf{w}_t - \mathbf{w}^*$ *satisfies*

$$\boldsymbol{\Delta}_{t+1}^T \mathbf{H}_t \boldsymbol{\Delta}_{t+1} \leq L\|\boldsymbol{\Delta}_{t+1}\|\|\boldsymbol{\Delta}_t\|^2 + \frac{\zeta_{comp,byz}^2}{1 - \zeta_{comp,byz}^2} \boldsymbol{\Delta}_t^T \mathbf{H}_t \boldsymbol{\Delta}_t + 2\epsilon_{comp,byz}^2.$$

572 *Proof.* We choose $\zeta = \zeta_{comp,byz}$ and $\epsilon = \epsilon_{comp,byz}$ from the Lemma 10 and follow the proof of
573 Lemma 6 to obtain the desired bound. $\qquad\square$

**Proof of Theorem 3**

575 *Proof.* We get the desired bound by developing from the result of the Lemma 11 and following the
576 proof of Theorem 1 $\qquad\square$

# 11 Additional Experiment

578 In addition to the experimental results in Section 6, we provide some more experiments supporting
579 the robustness of the COMRADE in two different types of attacks : 1. 'Gaussian attack': where the
580 Byzantine workers add Gaussian Noise $(\mathcal{N}(\mu, \sigma^2))$ to the update and 2. 'random label attack': where
581 the Byzantine worker machines learns based on random labels instead of proper labels.

(a) w5a 'Gauss'    (b) a9a 'Gauss'    (c) w5a 'random'    (d) a9a 'random'

(e) w5a 'Gauss'    (f) a9a 'Gauss'    (g) w5a 'random'    (h) a9a 'random'

Figure 3: (First row) Accuracy of COMRADE with $10\%, 15\%, 20\%$ Byzantine workers with 'Gaussian ' attack for (a). w5a (b). a9a and 'random label' attack for (c). w5a (d).a9a. (Second row) Accuracy of COMRADE with $\rho$-approximate compressor (Section 5) with $10\%, 15\%, 20\%$ Byzantine workers with 'Gaussian ' attack for (a). w5a (b). a9a and 'random label' attack for (c). w5a (d).a9a.