[Reviews · NeurIPS 2020]

Review 1

Summary and Contributions: The authors present a technique, called COMRADE, for distributed second-order optimization. In the technique, unlike previous techniques where Hessian and gradient are sent separately, each machine locally computes the Newton update (inverse Hessian * gradient) on its own points and sends it to the server. One contribution of the authors is to show that the mean of these local updates is still a good approximation of the Newton step. By filtering out the fraction of the updates with largest norm, the authors also make their scheme robust to a fraction of workers sending arbitrary updates. And if the updates are compressed, similar properties still hold, although slightly worse due to the compression. The authors provide expeiments showing their algorithm's performance on a logistic regression task.

Strengths: The three properties that the algorithm is shown to have (less communication, robustness, compressible) seem useful and advantageous, and distributed second-order optimization is obviously an important and realistic topic in machine learning. The authors make the case that in addressing robustness to Byzantine errors their work is novel. Based on the empirical experiments, performance seems to be good on a reasonable task.

Weaknesses: While my background is limited, it sounds like the general idea of having the workers locally compute updates may exist in previous algorithms, which could make the work somewhat less novel.

Correctness: The claims as stated make sense where I understand them, although I do not have the background to carefully check the proofs. The empirical methodology seems reasonable. While these are standard datasets and it is probably documented somewhere, it might help to include more details -- I especially wondered what the train/test split was, and what regularizer strength lambda was chosen.

Clarity: It is not a completely breezy read, as the work is very dense and compressed. It's a difficult topic in which I have limited background, and the paper is obviously very dense, but I was in the end able to understand it at least at a high level. I think this is a good sign for its clarity. There are grammar/usage errors and typos. It would be worth doing another pass to catch some of these. I would have found it helpful to see even very short proof sketches, if possible, in the main body of the work. Space is obviously limited, though.

Relation to Prior Work: Yes, multiple previous and related works are discussed, including one which is the baseline for empirical experiments, and the differences with those works are explained.

Reproducibility: Yes

Additional Feedback: After author response: Clarifying the experimental parameters is helpful. Since this is basically an optimization algorithm, reporting just training loss seems okay, but it couldn't hurt to have a train-test split too. My score remains the same.


Review 2

Summary and Contributions: In this paper, the authors propose a Byzantine-tolerant distributed Newton method which has small communication overhead. Theoretical analysis shows that the porposed algorithm COMRADE converges when the system is under Byzantine attacks on the workers.

Strengths: Byzantine-tolerant optimization is an interesting and important topic recently. I beleive this is the first work tackling Byzantine tolerance in Newton method.

Weaknesses: In the theoretical analysis, it seems that the error (epsilon) grows with m. Thus larger number of workers makes the convergence worse. At the end of Section 3, it seems that good convergence requires the local Hessian and gradient close to the global ones. Combined with the theoretical analysis, using distributed training might be meaningless. If the Hessians and gradients are similar to each other across the workers, then maybe it's unnecessary to use multiple Hessians and gradients. To justify the distributed training, in the experiments the authors should compare distributed COMRADE to Newton method with a single worker that only uses s samples (not all the n samples) in each step. There are other kinds of attacks the authors could try. For example, an attacker stronger than "negative" in this paper is that, the Byzantine workers (who are supposed to be omniscient) obtain the p from all the honest workers, take the average and then multiply the average with -c, where c \in (0,1) Byzantine workers could also add random noises to p.

Correctness: The paper is technically sound.

Clarity: The paper is well written and easy to follow.

Relation to Prior Work: The difference and improvement compared to the previous work is clearly discussed.

Reproducibility: Yes

Additional Feedback: --------------------- after authors' feedback The authors provided additional experiment result for other kinds of attacks. So I raise the score a little bit. Although I still have concerns about the error rate growing with the number of workers, which seems weakening the motivation of distributed training, after discussed with other reviewers I tend to accept the paper.


Review 3

Summary and Contributions: The paper considers a distributed second-order optimization algorithm and presents theoretical results on convergence rate. In addition, compression and byzantine resilience are explored. By using local gradients instead of the exact gradient, the proposed algorithm reduces the communication load between the workers and the master node by half at the expense of slower convergence.

Strengths: This paper presents convergence rates of the proposed distributed second order optimization under the different scenarios of compression and byzantine resilience. This paper discusses distributed second order optimization and this line of research is potentially important for the NeurIPS community.

Weaknesses: -Using local (sketched) gradients rather than the exact gradient is equivalent to “classical sketch” of [29] in a non-iterative algorithm, a discussion on this connection could be helpful. -There are likely some issues that need to be addressed regarding the experimental results. For instance in figure 1c, GIANT seems to converge to a lower accuracy than the proposed algorithm, this is probably because the step size was not tuned for the GIANT algorithm. I’d expect that both algorithms converge to the same accuracy. - All the plots in the experimental results section seem to be accuracy vs iteration plots. Since the paper presents an optimization algorithm, it could be helpful to include plots of loss vs iteration (or time). - Step size in algorithm 1 seems to be a fixed value, and there is no discussion on how to choose it. For instance, in the GIANT algorithm [30], the authors choose the step size using line search. Are the numerical results of the paper based on using a fixed step size? -The cost functions considered in the paper have regularization terms. However, it wasn’t mentioned how these have been chosen for the experiments. -Algorithm 1 runs for T iterations, how to choose T is not mentioned. Maybe the authors could give a stopping criterion. - There are some additional minor typos/inconsistencies throughout the paper. For instance, the equation for logistic loss in page 3 is 1-exp() instead of 1+exp(). Another one is that in the introduction, it is claimed that “we show that this sequential estimation is redundant” which might need editing since as it stands, it is not necessarily a correct claim. In figure 1e, only the green curve is visible.

Correctness: The theoretical results are probably correct. The empirical results could have some flaws.

Clarity: The paper is well written.

Relation to Prior Work: Yes

Reproducibility: Yes

Additional Feedback: I have read the rebuttal which addressed some of my concerns (regularization term and choosing T). However, the choice of the step size in the numerical experiments is not described in sufficient detail. I believe that competitor algorithm in Figure 1 can do better with step size tuning, which appears to be omitted. One can expect the full gradient based method to converge faster with the right step size. I agree with other reviewers that the convergence guarantee in Theorem 1 is weak. Besides the error floor increasing with the number of workers, there is dependence on the inverse of the minimum singular value and gradient norm, which can be quite large. Moreover, this is under a restrictive incoherency condition on the training data. In the response letter, the authors acknowledge that there are some direct connections to the randomized linear algebra literature. There are earlier model averaging algorithms proposed in the non-byzantine setting which are similar but not discussed. On the other hand, the simplicity of the norm based thresholding is an advantage. However it is not clear if this is the first time such a scheme was proposed in the Byzantine setting. For instance, one can apply the same strategy in a first order method. I will keep my score the same.


Review 4

Summary and Contributions: UPDATED: Thanks a lot for your answers! I keep my score at the same level. In my opinion, it is important to add the plots of 'loss vs. iterations' to the paper (from the optimization perspective, it seems to be more interesting than 'accuracy vs. iterations'). ================================ In this work, the authors present new second-order algorithm for solving empirical risk minimization problems with convex losses and l2 regularization. Their algorithm works in a distributed manner. At each iteration, the master sends to the workers the current point. Then, each worker processes its own 's' samples of data, building the corresponding gradient and the Hessian, and performing the classical Newton update. Finally, the workers send their updates to the master, which aggregates their results into the new point. The classical well-known result is the local quadratic convergence of the Newton method. The authors were able to extend this result for their setting. Their rate switches between local quadratic and local linear convergence, depending on the characteristics of the data matrix and sampling size 's'. Additionally, there is an additive error term, which suggests the maximum level of precision, which we can achieve by a concrete setup. Further, the authors extend their rates for the cases, when some of the workers are 'Byzantine', so the may communicate back damaged results. To tackle them, the master node performs some 'norm based thresholding'. Finally, the authors add a compressing operation for each node, to reduce the communication cost, and analyse the local convergence of the method under this most general regime. Numerical experiments are provided.

Strengths: 1. In my opinion, the main contribution of the paper is the analysis of the local converge of Newton method under different types of inexactnesses: * the effect of distributed parallel computations; * inexactness caused by Byzantine workers; * the effect of using a compression operation on the worker side. All these issues seem be important in practice. Provided analysis helps to determine the influence of these types of inexactnesses on the convergence of Newton method. 2. In numerical experiments, the proposed method outperforms another distributed second-order algorithm (GIANT) in terms of the number of iterations on the problem of training logistic regression model. The source code of the experiments is provided. 3. 'norm based thresholding' strategy for filtering the 'Byzantine' workers. Extensive experiments with different types of byzantine workers demonstrates really good performance.

Weaknesses: 1. The convergence result seems to be more 'qualitative', showing a general effect of different error sources, rather than 'quantitative'. Most of the terms have a complicated dependence on the parameters of the problem class. I think, they can hardly be computed in practice. For example, they did not provide a direct answer to the question: how to choose 's' in practice? It somewhat expected that for 'big enough 's', the stochastic approximation would be reasonably good. However, I still believe that the given theoretical answers may help to advance in developing distributed methods. 2. Numerical experiments. It is not clear what is the 'Accuracy' on the graphs (is it the functional residual?). It looks like all the methods stuck at some level (85-95). I think, it is important to demonstrate that in *some regimes* the method (for example, full Newton method) may achieve near 100 accuracy.

Correctness: The claims of the paper look correct. Though, I was not able to check all the details of the proofs.

Clarity: The paper is well written and easy to follow. It would be good to specify somewhere in Section 2, that lambda > 0 (the regularization parameter is strictly greater than zero), so the objective is strongly convex. Line 198: "It is well known that a distributed Newton method has linear-quadratic convergence rate." -- It would prefer to see some reference here. I do not fully understand Remark 3, if lambda > 0, then the Hessian is always invertible.

Relation to Prior Work: There is an extensive overview in Section 1. Comparing with state-of-the-art distributed second-order methods (lines 59-72), I would prefer to see more explanations, why communicating *once* per iteration is much better than *two times*. The difference by a constant factor does not seem to be significant, taking into account that second-order methods usually do not need to have so many iterations.

Reproducibility: Yes

Additional Feedback:

[Author Response · NeurIPS 2020]

We like to thank the reviewers for their insightful feedback. We answer the questions raised by them below.

**Reviewer 1:** We thank the reviewer for appreciating our work, and plan to add short proof sketches. Regarding
experiments, note that we only report the training accuracy of using the entire data. However, we have verified similar
performance via a $80 - 20$ spilt between train and test data. We choose regularization parameter to be 1.

**Reviewer 2** *"Error grows with the number of workers $m$"*: As seen in Theorem 1, the error ($\epsilon$) indeed is proportional
to $\sqrt{\log(m)}$, where $m$ is the number of workers. But, if $m$ increases, the error increases at a much lower rate. Moreover,
the error is also inversely proportional to $\sqrt{s}$, where $s$ is the number of samples at each machine. Usually, in practice, $s$
is much larger than $\log m$, and hence in this regime, the error remains small and our results are useful.
*"Unnecessary to use multiple Hessians and gradients"*: To verify the intuition about similarity between global and
local gradients, we ran the following experiment as per the reviewer's suggestion. Given a total of $n$ data points, we
allow each machine to sample $s$ data points. In Figure (a), we compare the convergence result (loss vs iteration) between
our setup with multiple worker nodes and single worker with sampled data (as the reviewer asked). It is evident from
the plot that our set up provides better result. For the single worker with exact Newton, the result is worse due to the
high variance as it is based on a fraction of the data. For our case the average of the local update reduce the variance and
provide better results. Hence, averaging the approximate update from local machines is better than the exact Newton
method in one machine.
*"Different Kinds of attacks"* We show the robustness of COMRADE when the byzantine machines send $-c \times p$ in
Figure (b), and $p + \mathcal{N}(0, 100)$ ('noise') in Figure (c), where we choose $c = 0.9$ and $p$ as the update only with the good
machines. It is evident that our algorithm is able to handle such byzantine attacks with norm based thresholding.

(a)        (b)        (c)        (d)

**Reviewer 3:** *"exact gradient is equivalent to classical sketch "*: In the special case of a squared loss, such connection
indeed exists, although we are considering more general loss functions. We will add a discussion on this.
*"GIANT seems to converge to a lower accuracy than the proposed algorithm"*: We believe that with rigorous tuning
of the parameters (like step size), both the algorithms will yield similar accuracy. In our paper, however, we choose
same parameter choices for both the algorithms. Our intent is to show even with one round communication, we can
match the accuracy of GIANT (which uses 2 rounds). In some experiments, owing to this fixed parameter setting,
COMRADE seems to achieve better accuracy.
*"how to choose T is not mentioned"* We run our algorithm, COMRADE for a sufficiently many iterations to ensure
convergence. Alternatively, for the stopping criteria we can also choose the norm of the update as an indicator. In
Figure (d), we plot the loss vs iteration with norm of the update ($\|p\|_2 < 0.1$) as a stopping criteria.
*"loss vs iteration plots":* Agreed. Note that in response to the previous question, we do provide a "loss vs iteration" plot.
*'Step size and line search':* We choose a fixed step size for all the experiments. GIANT uses Armijo–Goldstein
condition to choose the step size, which requires the knowledge of the full gradient. As the first round of communication
for the GIANT type algorithm computes the full gradient explicitly, the line search is feasible. In our case gradient
information is not available as our algorithm only computes local updates (Hessian inverse times the gradient based on
the local data). This provides the advantage of **less communication** but robs the opportunity to implement line search
type algorithm. But we show in our numerical results that we achieve good convergence even with fixed step-size.
*"The cost functions and regularization term"*: Apologies. In experiments, we choose regularization parameter to be 1.
*"Typos"* We will fix the typos and provide better visibility to plots.

**Reviewer 4** *"qualitative' rather than 'quantitative":* We agree with the assessment of the reviewer. Indeed the error
depends on several problem dependent parameters, and a clear choice of $s$ is difficult to obtain in practice. However,
our focus is to show that COMRADE converges to a vanishingly small error floor with sufficiently large $s$. We also
show that error decays in $\mathcal{O}(1/\sqrt{s})$ rate.
*"Numerical experiments":* By accuracy, we meant $1-$ the classification error (in percentage). With a rigorous tuning
of parameters and hyper-parameters, the algorithms may achieve close to $100\%$ accuracy. However, our goal is to show
that, for any fixed parameter setting, COMRADE (which communicates one round per iteration) can achieve the same
accuracy as the standard two round based algorithms such as GIANT.
*"$\lambda > 0$ and Remark 3":* Indeed, we only require $\lambda > 0$ to ensure the invertibility of the Hessian. We will rephrase the
remark. We will also add the strong convexity comment.
*"Linear-quadratic convergence reference"* Note that GIANT ([30] in the paper) has linear-quadratic rate of conver-
gence; also see [21] in the paper. We will add a few other references in the revised version as well.

[Meta-Review · NeurIPS 2020]

This work is a solid contribution to the literature on distributed optimization/training. Three reviewers suggested acceptance, and one suggested weak reject. I believe the concerns raised by this reviewer are not major and can (and should) be addressed in the camera ready version. I suggest acceptance provided the authors are ready to address all reasonable issues raised by the reviewers. Some more comments: Plus: There is a consensus that the paper's contributions are quite clear despite the dense content. I recommend some more details be added in the extra page available in the camera ready version. The simplicity of the norm based thresholding is a plus. Either state that this is the first time such thresholding is used to the best of your knowledge, or mention works where this was done before. Some issues: The choice of the step size in the numerical experiments is not described in sufficient detail. The competitor algorithm in Figure 1 can probably do better with step size tuning, which appears to be omitted. One can expect the full gradient based method to converge faster with the right step size. Please provide also plots of 'loss vs. iterations' in the revised version (from the optimization perspective, it seems to be much more interesting that 'accuracy vs. iterations'). There is a consensus among reviewers that the convergence guarantee in Theorem 1 is rather weak. Besides the error floor increasing with the number of workers, there is dependence on the inverse of the minimum singular value and gradient norm, which can be quite large. Moreover, this is under a restrictive incoherency condition on the training data. In their rebuttal, the authors acknowledge that there are some direct connections to the randomized linear algebra literature. I suggest you elaborate more on this in the camera ready. There are earlier model averaging algorithms proposed in the non-byzantine setting which are similar but not discussed. I suggest some coverage of these be included.